

# Coupled minimal models revisited II:
# Constraints from permutation symmetry

António Antunes[1,2]⋆ and Connor Behan[3]†

**1** Deutsches Elektronen-Synchrotron DESY, Notkestr. 85, 22607 Hamburg, Germany
**2** Laboratoire de Physique, École Normale Supérieure, Université PSL, CNRS,
Sorbonne Université, Université Paris Cité, 24 rue Lhomond, F-75005 Paris, France
**3** ICTP South American Institute for Fundamental Research, Instituto de Física Teórica
UNESP, Rua Dr. Bento Teobaldo Ferraz 271, 01140-070, São Paulo, SP, Brazil

⋆ antonio.antunes@phys.ens.fr , † connor.behan@gmail.com

## Abstract

Coupling $N$ large $m$ minimal models and flowing to IR fixed points is a systematic way to build new classes of compact unitary 2d CFTs which are likely to be irrational, and potentially have a positive Virasoro twist gap above the vacuum. In this paper, we build on the construction of [1], establishing that, for spins less than 10, additional currents transforming in non-trivial irreducible representations of the permutation symmetry $S_N$ are not conserved at the IR fixed points. Along the way, we develop a finer understanding of the spectrum of these theories, of the special properties of the $N = 4$ case and of non-invertible symmetries that constrain them. We also discuss variations of the original setup of [1], some of which can exist for smaller values of the UV central charge.



# 1   Introduction

Two-dimensional conformal field theories (CFTs) play a privileged role in the space of quantum field theories (QFTs). Among them, there are examples of interacting QFTs whose spectrum [2] and correlation functions of local operators [3,4] and even of extended operators [5,6] can be computed exactly.[1] Exactly solvable *unitary* 2d CFTs fall into two main classes of examples:[2]

- *Rational* CFTs, which by definition have a finite number of primary operators of their chiral symmetry algebra, and are in particular *compact*, meaning they have a discrete spectrum. The archetypical examples of such theories are the Virasoro minimal models.

- Non-compact irrational CFTs, i.e. CFTs with a continuous spectrum of primaries. The key example here is the Liouville family of CFTs parametrized by the central charge $c$ [10,11].

Instead of discussing the many interesting generalizations of these two classes of exactly solvable models, we want to ask about the missing middle:

> *Q1: What is the space of compact irrational CFTs?*

Theories with a discrete but infinite set of primary operators span the known space of unitary CFTs in higher dimensions. Instead, in two dimensions, they are sometimes portrayed as generic and other times as esoteric, ill-defined objects, much like transcendental numbers on the real line. One class of compact irrational CFTs can at least easily be proved to be so, even if their spectrum and correlation functions are hard to determine: theories that lie in the same conformal manifold as an RCFT. In these cases we can make use of Vafa's theorem [12]: RCFTs have scaling dimensions and central charges that are rational numbers. In a conformal manifold, starting from a rational point, as long as any scaling dimension changes continuously, it is guaranteed to take irrational values somewhere. The compact boson at an irrational radius and supersymmetric non-linear sigma models at generic points on the moduli space fall into this class.[3] Of course, these models are either free or supersymmetric and in the spirit of genericity we would like to refine our question to

> *Q2: What is the space of interacting compact irrational CFTs with no extended chiral symmetry?*

---

[1]Supersymmetric QFTs (including in higher spacetime dimensions) also possess certain observables that can be computed exactly but none of them can be solved to the extent of 2d CFTs.

[2]See however the work of [7–9] for progress in the solution of loop model families of non-unitary 2d CFTs.

[3]The authors of [13] showed that irrational central charges also arise from a generalization of the Sugawara construction. This suggestion has not led to any concrete model since the action of $L_0 = \sum_{m>0} \ell_{ab} J^a_{-m} J^b_m$ on a module of the current algebra is hard to diagonalize when $\ell_{ab}$ is not a Killing form.

In an attempt to answer this question, in [1] we investigated infrared (IR) fixed points of $N$ coupled minimal models preserving a permutation symmetry $S_N$. This is an old construction, that was often considered in the context of disordered systems and the replica trick, leading to a replica symmetry $S_N$, where one is ultimately interested in the $N \to 0$ limit. These studies typically involved coupled Ising and $q$-state Potts models [14–17], and made significant use of the $(q-2)$ expansion. In particular, the authors of [18], using several numerical methods including Monte Carlo and transfer matrix techniques, established the existence of an infrared fixed point for $N = 3$ coupled 3-state Potts models with a central charge $c \approx 2.377$. This is incompatible with known RCFTs with the correct symmetry, suggesting that it is a simple example of an isolated compact irrational CFT.[4] The precise coupling is

$$S = \sum_{i=1}^{3} S_{q-\text{Potts}}^{i} + g \int d^2x \sum_{i<j} \epsilon^i \epsilon^j \,. \tag{1}$$

As discussed in [20], CFT data in ultraviolet (UV) theory is conveniently specified in terms of a coupling constant $f$ such that $q = 2 + 2\cos(\pi f/2)$. The critical and tricritical $q$-state Potts models, which annihilate at $q = 4$, are described in the range $q \in [0,4]$ by the branches $f \in [2,4]$ and $f \in [4,6]$ respectively. In this notation, $\epsilon^i$ starts off with the holomorphic dimension $\frac{3}{f} - \frac{1}{2}$ which means $\epsilon^i \epsilon^j$ is only classically marginal for $q = 2$. Although the tensor product CFT in (1) is solvable and unitary for $q = 3$, the renormalization group (RG) flow away from it is strongly coupled. To make it weakly coupled, we must give up unitarity by taking $q$ to be close to 2.

The construction of [1] extends the range of potential unitary infrared CFTs by instead coupling minimal models indexed by $m$ and using $1/m$ as an expansion parameter, leading to infinite sequences of unitary weakly coupled fixed points as $1/m$ approaches 0.[5] A key property of these fixed points is that additional $S_N$ singlet currents of the UV tensor product theory (checked up to spin $J \leq 10$) stop being conserved in the infrared. Evidence for the breaking of chiral symmetry is, at the same time, evidence for irrationality because these models are modular invariant with $c > 1$. The growth estimates in [21,22], showing that there are infinitely many Virasoro primaries, are therefore applicable meaning that an RCFT is not possible without an extended chiral algebra. In this work, we extend the results of [1] in several directions:

1. We emphasize that analyzing singlet currents is sufficient to establish irrationality when gauging the $S_N$ symmetry of the original model.

2. When the symmetry is ungauged, making use of refined partition functions, we classify the currents in all non-trivial representations of $S_N$ and show that those below spin 10 acquire anomalous dimensions. We further argue that this should remain true at large but finite $m$.

3. For the $N = 4$ system, we show that it preserves a large class of non-invertible symmetries and give a non-perturbative argument for the integrability of an associated deformation.

4. We construct other systems of coupled minimal models which additionally include some copies at fixed $m$, and can even exist for central charges smaller than the model of [18].

In broad strokes, this work demonstrates the importance of being able to systematically search a large space of operators which are computationally intensive to construct. Lessons from this approach have already become apparent in other contexts, including [23] which

---

[4]See [19] for a recent attempt to bootstrap this model.

[5]In the convention we use, the first unitary minimal model (the critical Ising model) corresponds to $m = 3$.

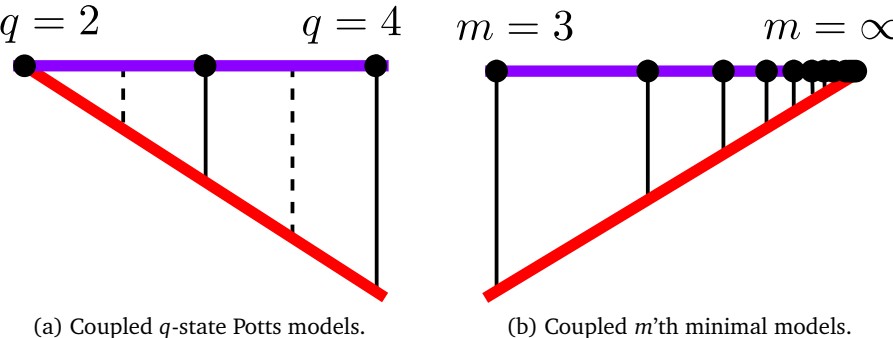

(a) Coupled $q$-state Potts models.  (b) Coupled $m$'th minimal models.

Figure 1: Schematic diagrams of RG flows from the UV (violet) to IR (red). The ones written with dashed lines are non-unitary.

revisited the problem of describing black hole microstates. In the present case, an algorithmic search has led to strong evidence that the models we proposed are irrational CFTs with chiral symmetry given by a single copy of the Virasoro algebra. Although the extra symmetry is broken very weakly in the perturbative regime studied here, the fact that it is broken at all opens the door to various types of numerical simulation, some of which have had recent success at observing chaos at high energies [24].[6] We expect that the ability to find similar models at smaller values of the central charge will be helpful for making this route more accessible.

This paper is organized as follows. In section 2, we review the $S_N$ symmetric models of [1] and the technique that was used to decide between conservation and non-conservation of the singlet UV currents. In particular, we advocate for a more thorough check which goes beyond the $S_N$ singlet sector. Next, in section 3, we sharpen our expectations regarding the non-singlet sector with the use of torus partition functions and $S_N$ representation theory. Section 4 then uses this understanding to develop algorithms which are able to perform the desired more powerful check. This leads to the result that there are still no signs of any chiral symmetry beyond Virasoro. Extensions of the models studied here along with an analysis of non-invertible symmetries in both the original and extended models are discussed in section 5 before we give an outlook on future directions in section 6.

## 2 Review of singlets

### 2.1 The model

The models considered in [1], which this paper will study in more detail, have been designed to circumvent the problem with the left diagram of Figure 1 in favour of the right diagram. There are many tensor products of solvable CFTs which allow one to construct weakly coupled RG flows to less familiar fixed points. But simply constructing flows which become arbitrarily short is not enough. The theories from which they emanate must also be unitary if one is to have a hope of reaching the physically interesting case of a compact unitary irrational CFT with minimal chiral symmetry. This makes it quite natural to take minimal models as our seed theories and work in the regime where unitary minimal models accumulate, namely $m \to \infty$. This gives us access to large-$m$ perturbation theory which was first used in [26,27] and later in [28,29] for different purposes.

---

[6]See also [25] for recent progress on the understanding of chaos in 2d CFTs.

To fix notation, the unitary diagonal minimal model labeled by the integer $m \geq 3$ has central charge

$$c = 1 - \frac{6}{m(m+1)}, \tag{2}$$

and finitely many Virasoro primaries $\phi_{(r,s)}$. These have the weights $h = \bar{h} = h_{(r,s)}$ for

$$h_{(r,s)} = \frac{[(m+1)r - ms]^2 - 1}{4m(m+1)} = \frac{(r-s)^2}{4} + \frac{r^2 - s^2}{4m} + \frac{s^2 - 1}{4m^2} + O(m^{-3}), \tag{3}$$

and integer $(r,s)$. To be precise, $\phi_{(r,s)}$ is identified with $\phi_{(m-r,m+1-s)}$ and $(r,s)$ is in the so called Kac table $[1, m-1] \times [1, m]$. Analytically continuing in $m$ leads to the concept of a generalized minimal model [30, 31] (see also [32, 33]). Generalized minimal models are labeled by a parameter $b \in \mathbb{C}$ and have a central charge and spectrum agreeing with (2) and (3) when $b^2 = -\frac{m+1}{m}$.[7] Minimal model structure constants $C_{(r_1,s_1)(r_2,s_2)(r_3,s_3)}$ can be recovered as well if we take $r_i, s_i \ll m$ but otherwise this is a subtle issue [34]. To see why, consider the truncation of the fusion rules which occurs in a unitary diagonal minimal model.

$$
\begin{aligned}
\phi_{(r_1,s_1)} \times \phi_{(r_2,s_2)} &= \sum_{r_3=|r_1-r_2|+1}^{r_1+r_2-1} \sum_{s_3=|s_1-s_2|+1}^{s_1+s_2-1} \phi_{(r_3,s_3)} \\
&\rightarrow \sum_{r_3=|r_1-r_2|+1}^{\min(r_1+r_2, 2m-r_1-r_2)-1} \sum_{s_3=|s_1-s_2|+1}^{\min(s_1+s_2, 2m+2-s_1-s_2)-1} \phi_{(r_3,s_3)}.
\end{aligned}
\tag{4}
$$

This states that minimal model structure constants vanish when $r_0 \geq m$ or $s_0 \geq m+1$ for

$$r_0 = \frac{r_1 + r_2 + r_3 - 1}{2}, \qquad s_0 = \frac{s_1 + s_2 + s_3 - 1}{2}. \tag{5}$$

Conversely, the generalized minimal model structure constants do not always have this property. In terms of

$$
\begin{aligned}
P_r(x) &= \prod_{i=1}^{r} \frac{\Gamma(1+ix)}{\Gamma(-ix)}, \\
Q_{r,s}(b) &= \prod_{i=1}^{r} \prod_{j=1}^{s} (ib + jb^{-1})^2, \\
R_{r,s}(b) &= \frac{\Gamma(-rb^2 - s)\Gamma(-r - sb^{-2})}{\Gamma(rb^2 + s)\Gamma(r + sb^{-2})},
\end{aligned}
\tag{6}
$$

they are presented as

$$
\begin{aligned}
C_{(r_1,s_1)(r_2,s_2)(r_3,s_3)} &= \sqrt{\frac{R_{1,1}(b)}{\prod_{i=1}^{3} R_{r_i,s_i}(b)}} \frac{(-b^2)^{r_0-s_0} f_{r_1,r_2,r_3} f_{s_1,s_2,s_3}}{P_{r_0}(b^2) P_{s_0}(b^{-2}) Q_{r_0,s_0}(b)} \\
&\times \prod_{i=1}^{3} \frac{P_{r_i-1}(b^2) P_{s_i-1}(b^{-2}) Q_{r_i-1,s_i-1}(b)}{P_{r_0-r_i}(b^2) P_{s_0-s_i}(b^{-2}) Q_{r_0-r_i,s_0-s_i}(b)},
\end{aligned}
\tag{7}
$$

in [32].[8] Here, $f_{r_1,r_2,r_3}$ and $f_{s_1,s_2,s_3}$ are 1 when the non-truncated fusion rules (first line of (4)) are obeyed and 0 otherwise. To check whether we also recover the truncated fusion rules, it

---

[7]If one simply wants a minimal model (finitely many primaries) but does not care about unitarity, $b^2 = -\frac{p}{p'}$ for relatively prime integers $p$ and $p'$ is also an option.

[8]Although we will not do so here, it is common to absorb the square root into two-point functions instead so that all correlation functions are meromorphic functions of the central charge.

Table 1: The behaviour of denominator factors in (7) for all four copies of the Kac table that can be reached by $(r_0, s_0)$. Minimal models have vanishing structure constants in all but the upper left copy. Generalized minimal models, on the other hand, can have them be non-vanishing in the lower right copy due to the cancellation between $Q_{r_0,s_0}(b)$ and $P_{r_0}(b^2)P_{s_0}(b^{-2})$.

|  | $r_0 < m$ | $r_0 \geq m$ |
|---|---|---|
| $s_0 < m + 1$ |  | $P_{r_0}(b^2) \to \infty$ |
| $s_0 \geq m + 1$ | $P_{s_0}(b^{-2}) \to \infty$ | $P_{r_0}(b^2), P_{s_0}(b^{-2}) \to \infty, Q_{r_0,s_0}(b) \to 0$ |

helps to notice that $r_i < m \Rightarrow r_0 - r_1 < m$ and $s_i < m + 1 \Rightarrow s_0 - s_i < m + 1$. This means all factors in the second line of (7) are innocuous. Looking at the first line however, $(r_i, s_i)$ being in the Kac table only restricts $(r_0, s_0)$ to the first four copies of the Kac table. Table 1 then shows that minimal model and generalized minimal model structure constants disagree in one of these copies.[9]

Having committed to $m \gg 1$, in order to have both a sensible analytic continuation and perturbative control, we should identify relevant operators which can be used to initiate a flow. This can be done in three steps.

1. According to (3), $\phi_{(r,r)}$, $\phi_{(r,r+1)}$, $\phi_{(r,r+2)}$ and $\phi_{(r,r-1)}$ are all relevant.

2. To get a weakly relevant operator by coupling more than one model, there should be an integer multiple of $h_{(r,s)}$ which is slightly less than 1 instead of slightly more. This narrows the above list down to $\phi_{(r,r+1)}$ and $\phi_{(r,r+2)}$.

3. In order to have perturbative control, it is also important to work with a finite set of relevant operators which does not generate new ones under repeated OPEs. According to (4), this finally narrows the list down to $\phi_{(1,2)}$ and $\phi_{(1,3)}$.

It is now clear that we should take a tensor product of $N \geq 4$ minimal models and deform using a sum of $\phi_{(1,3)}$ operators and a four-fold product of $\phi_{(1,2)}$. Demanding $S_N$ symmetry leads to the formal action[10]

$$S_{\text{CMM}} = \sum_{i=1}^{N} S_m^i + \int d^2x \left( g_\sigma \sigma + g_\epsilon \epsilon \right), \tag{8}$$

where

$$\sigma \equiv \binom{N}{4}^{-\frac{1}{2}} \sum_{i<j<k<l}^{N} \phi_{(1,2)}^i \phi_{(1,2)}^j \phi_{(1,2)}^k \phi_{(1,2)}^l, \qquad \epsilon \equiv N^{-\frac{1}{2}} \sum_{i=1}^{N} \phi_{(1,3)}^i. \tag{9}$$

We have taken all $\phi_{(r,s)}$ to be unit-normalized so that the Zamolodchikov metric,

$$\mathcal{N}_{IJ} \equiv \left\langle \mathcal{O}_I(0) \mathcal{O}_j(\infty) \right\rangle, \tag{10}$$

is trivial. This makes $C_{(r_1,s_1)(r_2,s_2)}^{(r_3,s_3)}$, which is what naturally enters in conformal perturbation theory, the same as $C_{(r_1,s_1)(r_2,s_2)(r_3,s_3)}$ from (7). The cases we will need are

$$C_{(1,2)(1,2)}^{(1,3)} = \frac{\sqrt{3}}{2} + O(m^{-1}), \qquad C_{(1,3)(1,3)}^{(1,3)} = \frac{4}{\sqrt{3}} + O(m^{-1}). \tag{11}$$

---

[9]For non-diagonal minimal models in the D-series, there is actually an analytic continuation which does not have this issue but on the other hand it is non-compact [35].

[10]This action was first considered by [16] in the context of disordered models. Unfortunately, we only became aware of this work (which appears to be almost uncited) after publication of [1].

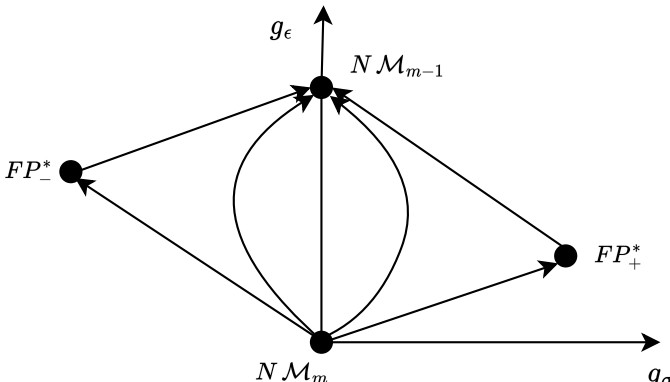

Figure 2: Schematic diagram of the fixed points reproduced from [1]. The tricritical fixed points $FP^*_\pm$ given by (14) are the conjectured examples of compact unitary irrational CFTs with only Virasoro symmetry.

A leading order analysis in conformal perturbation theory [36] is now straightforward. For the deformation $\int d^2x\, g^I \mathcal{O}_I$, we will define $\tilde{g}^I \equiv (1-h_I)g^I$ so that the summation convention may be used in the well known expressions for the beta-function and the c-function in [37,38]. They read

$$
\begin{aligned}
\beta^I &= 2\tilde{g}^I - \pi C^I_{JK} g^J g^K + O(g^3), \\
\Delta c &= -2\pi^2 \mathcal{N}_{IJ} g^J (3\tilde{g}^I - \pi C^I_{KL} g^K g^L) + O(g^4).
\end{aligned}
\tag{12}
$$

After computing three-point functions in the tensor product theory, the first line of (12) becomes

$$
\begin{aligned}
\beta_\sigma &= \frac{6}{m} g_\sigma - \frac{4\pi\sqrt{3}}{\sqrt{N}} g_\sigma g_\epsilon - 6\pi \binom{N-4}{2}\binom{N}{4}^{-\frac{1}{2}} g_\sigma^2, \\
\beta_\epsilon &= \frac{4}{m} g_\epsilon - \frac{4\pi}{\sqrt{3N}} g_\epsilon^2 - \frac{2\pi\sqrt{3}}{\sqrt{N}} g_\sigma^2,
\end{aligned}
\tag{13}
$$

up to $O(g^3)$ terms. Setting (13) to zero leads to four fixed points. The obvious ones are $(g_\sigma^*, g_\epsilon^*) = (0,0)$ which is fully unstable and $(g_\sigma^*, g_\epsilon^*) = \left(0, \frac{\sqrt{3N}}{m\pi}\right)$ which is fully stable (following immediately from the fact that it is a decoupled fixed point).[11] The latter is nothing but $N$ copies of the minimal model with $m$ replaced by $m-1$ [26, 27]. The more interesting fixed points $FP^*_\pm$ are

$$
g^*_{\sigma\pm} = \pm\frac{\sqrt{(N-3)_4}}{\pi m \sqrt{2P(N)}}, \qquad g^*_{\epsilon\pm} = \frac{\pm Q(N) + \sqrt{3P(N)}}{2\pi m \sqrt{P(N)/N}},
\tag{14}
$$

with

$$
P(N) = 3N^4 - 53N^3 + 357N^2 - 1069N + 1194, \qquad Q(N) = 3N^2 - 27N + 60.
\tag{15}
$$

These have one stable and one unstable direction, corresponding to one positive and one negative eigenvalue of $\frac{\partial \beta^I}{\partial g^J}$. The picture that emerges is Figure 2.

At this point, some related studies like [28] have recalled Vafa's theorem [12] which states that the modular $T$-matrix acting on the characters of a rational CFT must have finite order.

---

[11]Stable and unstable here are being used in the RG sense. It is expected that all coupled minimal model fixed points have a stable vacuum because they are not perturbations around a free theory.

As recalled in the introduction, this implies that the central charge and all conformal weights are rational numbers. Motivated by this result, [1] pointed out that irrational numbers appear in the leading order results for $\Delta c$ and the eigenvalues of $\frac{\partial \beta^I}{\partial g^J}$. Without strong assumptions about the $FP^*_\pm$ fixed points belonging to analytic families, this curiosity cannot hope to show that the CFTs are irrational because it ignores higher loop effects.

## 2.2 Anomalous dimensions of singlet currents

To find evidence of irrationality which is actually convincing, [1] undertook a search for generators of an extended chiral algebra in the IR and found none. Due to its large $\mathfrak{Vir}^N$ chiral algebra, the UV has many such currents but none of them were found to still be conserved in the IR. Specifically, this check was done for spin $J \le 10$ currents in the singlet representation of $S_N$ which transform as primaries under the diagonal $\widehat{\mathfrak{Vir}}$ generated by the sum of stress tensors. Since $\widehat{\mathfrak{Vir}}$ remains unbroken in the IR, local operators must arrange themselves into representations with respect to this symmetry. Within a given multiplet, the anomalous dimensions (if any) are the same for the primaries and descendants so it is sufficient to only check the former. On the other hand, the singlet assumption will be relaxed very soon.

When a UV current transforming as a primary under $\widehat{\mathfrak{Vir}}$ fails to be conserved in the IR, this means that its anomalous dimension is non-zero. Computing this anomalous dimension directly requires conformal perturbation theory at two loops. Indeed, $\left\langle T^K \mathcal{O} T^L \right\rangle = 0$ whenever $T^K$ is a spin-$J$ current and $\mathcal{O} \in \{\sigma, \epsilon\}$ is a deforming operator. Fortunately, [1] was able to avoid a two-loop calculation by using a more modern method which exploits the non-conservation equation

$$\bar{\partial} T^K = \bar{b}(g_\sigma)^K_L V^L, \qquad \bar{b}(g_\sigma)^K_L = b^K_L g_\sigma + O(g_\sigma^2), \tag{16}$$

and its consequences for conformal multiplets. This method, which we will now describe, was developed in [39] and has since become standard in the study of weakly broken higher spin symmetries [40–58].

The initial observation is that when the operator content of the theory changes continuously, taking $g_\sigma \to 0$ in (16) leaves behind a short multiplet with $(h, \bar{h}) = (J, 0)$ and a long multiplet with $(h, \bar{h}) = (J, 1)$. For a given spin $J$, it is conceptually straightforward to enumerate a basis of $(h, \bar{h}) = (J, 0)$ operators whose elements will be denoted $T^K$. These are what we have been referring to as *(UV) currents*. It is similarly possible to produce a basis for the space of $(h, \bar{h}) = (J, 1)$ operators with elements $V^K$. In practice, one should regard these bases as arbitrary which means they are almost certainly not nice enough to yield $T^K$ and $V^K$ operators which match up one-to-one. As such, the divergence of a given $T^K$ at one loop is a non-trivial linear combination of the $V^K$ whose coefficients are encoded in the all-important matrix $b^K_L$. Before this linear combination is taken, the $V^K$ will be referred to as *divergence candidates* of a spin-$J$ current.[12]

The multiplets described above must recombine when the coupling $g_\sigma$ is turned on and this leads to a concrete algorithm for determining which currents lift. This revolves around computing $b^K_L$ which yields the precise linear combination of divergence candidates that enters in (16). It also fixes the anomalous dimension $\gamma$ of the broken current since

$$g_\sigma^2 b^K_L b^{K'}_{L'} \left\langle V^L(z_1, \bar{z}_1) V^{L'}(z_2, \bar{z}_2) \right\rangle = \left\langle \bar{\partial} T^K(z_1, \bar{z}_1) \bar{\partial} T^{K'}(z_2, \bar{z}_2) \right\rangle$$

$$= \bar{\partial}_1 \bar{\partial}_2 \frac{\delta^{KK'}}{z_{12}^{2J+\gamma} \bar{z}_{12}^\gamma}. \tag{17}$$

---

[12] Since we are studying the lifting induced by $\sigma$, one more condition we will often impose is that four copies of $\phi_{(1,2)}$ must be present in the expression for $V^K$ to make it a divergence candidate.

In order to find the matrix $b^K_L$, we can use the fact that the first perturbative correction to $\left\langle \bar{\partial} T^K(z_1) V^L(z_2, \bar{z}_2) \right\rangle$ is

$$g_\sigma \int d^2 z_3 \left\langle \bar{\partial} T^K(z_1) V^L(z_2, \bar{z}_2) \sigma(z_3, \bar{z}_3) \right\rangle, \tag{18}$$

where the integrand is a three-point function in the undeformed theory. On the other hand, (16) turns the same correlator into

$$g_\sigma b^K_{L'} \left\langle V^{L'}(z_1, \bar{z}_1) V^L(z_2, \bar{z}_2) \right\rangle = g_\sigma b^K_L \frac{1}{z_{12}^{2J} \bar{z}_{12}^2}. \tag{19}$$

Setting (18) equal to (19), we can solve for $b^K_L$ by performing a calculation at one-loop instead of two. Moreover, the three-point function at zero loops that we are instructed to integrate vanishes at separated points due to $\bar{\partial} T^K(z_1)$. This leads to the appearance of a delta function which makes the integral trivial, yielding the relation

$$b^K_L = \pi C^K_{L\sigma} + O(1/m), \tag{20}$$

where $C^K_{L\sigma}$ is the OPE coefficient for $\left\langle T^K(z_1) V^L(z_2, \bar{z}_2) \sigma(z_3, \bar{z}_3) \right\rangle$. Due to the simplification of this quantity, one can observe a key fact which is that the only way for $T^K$ to stay conserved is to have $b^K_L = C^K_{L\sigma} = 0$ for all $L$. While $b^K_L \neq 0$ for a single $L$ is sufficient to conclude that $T^K$ is not a dilation eigenstate with weights of $(h, \bar{h}) = (J, 0)$ in the IR, we are more interested in establishing this for arbitrary linear combinations of the $T^K$. As such, the check to undertake is that $b^K_L$ is a matrix whose rank is equal to the number of rows. This is what led to the conclusion in [1] that $S_N$ singlets up to spin-10 for $N = 5, 6, 7$ all lift. The next two sections are about applying the same strategy to non-singlets.

## 2.3  When are singlets enough?

The restriction to singlets in [1] was motivated by the following fact. Whenever the IR fixed point has a chiral operator $T^K$ in some representation of $S_N$, normal ordered products of $T^K$ contracted with invariant tensors are necessarily chiral as well. This means that if singlet currents do not exist up to $J = 10$ then arbitrary currents do not exist up to $J = 5$. The problem with this argument of course is that singlet currents do exist up to $J = 10$ — they are just in the $\widehat{\mathfrak{Vir}}$ identity multiplet. A search for non-singlet currents, going beyond [1], is therefore preferable for two reasons. First, the bound $J = 5$ is not particularly high and one would like to go higher. Second, there is a "finely tuned" way to have non-singlet currents even below $J = 5$ and it needs to be checked whether or not coupled minimal model fixed points take advantage of this mechanism.

On the other hand, there is a related setup in which the search for singlet currents can be considered complete. Instead of deforming a tensor product of minimal models as in (8), one can instead deform their symmetric product orbifold. As reviewed in [59], states of the original tensor product theory which survive the orbifold (or discrete gauging) procedure are the ones invariant under $S_N$ permutations. In addition to this, new states are included so that the resulting torus partition function can still be modular invariant. These are $S_N$ invariant states with twisted boundary conditions along the spatial circle. There is one twisted sector for each conjugacy class, namely each choice of non-negative integers $(l_k)$ such that $\sum_{k=1}^N k l_k = N$ [60]. Within a given sector,

$$h, \bar{h} \geq \frac{c}{24} \sum_{k=1}^N l_k \left( k - \frac{1}{k} \right), \tag{21}$$

where $c$ is the single-copy central charge in (2). In particular, all conserved currents are in the untwisted sector and therefore coincide with the ones that were examined in [1]. An important point is that the gauging described here fails for symmetries that have a 't Hooft anomaly due to an ambiguity in how one prepares gauge invariant states with twisted boundary conditions. As explained in [61], one can diagnose this by looking for spins $J \notin \frac{\mathbb{Z}}{2}$ in the modular images of the partition function with a topological defect line inserted [62]. This can happen for any global symmetry $G$ with non-trivial $H^3(G, U(1))$ but not for a permutation symmetry which acts between copies. Indeed, there is a formula

$$\sum_{N=0}^{\infty} t^N Z_{S_N}(\tau, \bar{\tau}) = \exp\left[\sum_{k=1}^{\infty} t^k T_k Z(\tau, \bar{\tau})\right], \tag{22}$$

due to [63] which computes the partition function of any symmetric product orbifold using Hecke operators which act on the partition function of the seed theory according to

$$T_k Z(\tau, \bar{\tau}) = \sum_{d|k, a=\frac{k}{d}} \sum_{b=0}^{d-1} Z\left(\frac{a\tau+b}{d}, \frac{a\bar{\tau}+b}{d}\right). \tag{23}$$

The defining property of Hecke operators now ensures that modular invariance of $Z_{S_N}(\tau, \bar{\tau})$ follows immediately from modular invariance of $Z(\tau, \bar{\tau})$.[13] The orbifold by $S_N$ is therefore well defined for any tensor product CFT.

Before developing technology for non-singlet currents, which will be important for the non-orbifold theory, let us point out that conformal perturbation theory for symmetric orbifolds has recently come to play a large role in studies of $AdS_3/CFT_2$. In the most well established example [64, 65], the seed theory is a sigma model for $K3$ or $T^4$ and the subsequent deformation is by an exactly marginal operator in the twisted sector. In this case, higher-spin currents (and other short multiplets) are again lifted but the deformed theories have more than just Virasoro symmetry since they are supersymmetric. Some work in this direction, including the identification of Regge trajectories, has been done in [66, 67]. A less supersymmetric version of $AdS_3/CFT_2$ using $\mathcal{N} = 2$ minimal models, which has been proposed in [68–70], also exhibits lifting of higher-spin currents as one moves toward irrational SCFTs on the moduli space. A possible inverse to this phenomenon, wherein one encounters new rational points, has been studied perturbatively in [71, 72].

# 3 Permutation symmetry and charged currents

The technique in section 2.2, which alleviates the need for direct two-loop conformal perturbation theory, makes it crucial to understand the spaces of both UV currents and the operators with the right quantum numbers to recombine with them. Virasoro characters provide an efficient starting point for this task. In particular, they allow us to obtain the dimension of either space for arbitrarily high spins and to refine this dimension by $S_N$ representations. This section will explain the steps involved focusing mostly, but not exclusively, on the large $m$ regime where perturbative computations are most reliable.

## 3.1 Crash-course in $S_N$ representation theory

The systems of coupled models we work with exhibit a discrete global symmetry $G \supset S_N$ along with extra $\mathbb{Z}_2$ factors depending on whether $m$ is even or odd and whether $N$ is equal

---

[13]To see that $T_k Z(\tau, \bar{\tau})$ is modular invariant, note that (23) is a sum over upper triangular matrices $M$ with determinant $k$. When computing $T_k Z(\gamma\tau, \gamma\bar{\tau})$ for $\gamma \in SL(2, \mathbb{Z})$, $M\gamma$ is no longer upper triangular but it can be brought back to this form by acting with another $SL(2, \mathbb{Z})$ matrix on the left. This preserves $Z(\tau, \bar{\tau})$ by assumption.

to or larger than 4. For all cases, the permutation symmetry is present and therefore the spectrum organizes itself into irreducible representations of $S_N$. Since part of our work involves explicitly constructing operators transforming in these representations, we will quickly review their structure for convenience of the reader. A useful reference is [73].

Irreducible representations of $S_N$ are in one-to-one correspondence with integer partitions of $N$ which can be denoted either by a tuple $\lambda = (\lambda_1, \ldots, \lambda_N)$ with $\lambda_1 \geq \lambda_2 \geq \cdots \geq \lambda_N$ and $\sum_i \lambda_i = N$ or by a Young diagram with at most $N$ rows and a non-increasing number of left-justified boxes in each row. The dimension of each representation is given by the hook length formula

$$\dim(\lambda) = \frac{N!}{\prod_{i,j} h_\lambda(i,j)}, \tag{24}$$

where the product runs over $i, j$ which label the cells of the Young diagram, and for each cell one computes the hook length $h_\lambda(i,j)$ which is given by the number of cells to the right plus the number of cells below $(i,j)$, counting the cell itself once.

Let us illustrate this by considering simple representations which appear universally in the models of interest and determining their dimension. Consider first the trivial partition $(N)$ corresponding to the Young diagram with $N$ boxes in a single row

$$(N) = \boxed{\phantom{x}\phantom{x}\boxed{\ldots}\phantom{x}}. \tag{25}$$

Clearly the hook length formula gives $\dim(N) = 1$, and this is just the singlet representation. The next-to-simplest partition is given by

$$(N-1, 1) = \boxed{\phantom{xx}}. \tag{26}$$

Now the hook formula gives $\dim(N-1, 1) = N-1$. This is the so-called standard representation of $S_N$. This representation can be realized by any object transforming as a vector of $O(N)$, say $X^i$ with $i = 1, \ldots, N$, subject to the $S_N$ traceless condition $\sum_i X^i = 0$, which eliminates the degree of freedom associated to the singlet $\sum_i X^i$. This simply means that the fundamental representation of $O(N)$ decomposes into a standard representation and a singlet of $S_N$. This can be formulated in terms of an additional $S_N$ invariant tensor $e_i = (1\,1\ldots 1)$, which can be used remove additional traces. With this mind, we will often think of $S_N$ irreps as $O(N)$ tensors, which are labeled by the Young tableau where the top row is removed, since this still determines the partition uniquely. The relation between $S_N$ and $O(N)$ irreps is often known as Schur-Weyl duality.

In the tensor product theory, we build operators/states by multiplying building blocks which carry a fundamental index $i = 1, \ldots, N$, for example, we can build the two index object

$$L^i_{-2} L^j_{-2} |0\rangle, \tag{27}$$

which we need to understand how to decompose into irreducible representations. More generally, we want to determine how tensor products of irreducible representations decompose into direct sums of irreducible representations. For our purposes, it is sufficient to determine how to take $n$-fold tensor products of the standard representation. Indeed, for the state above, we need to determine

$$((N) \oplus (N-1, 1)) \otimes ((N) \oplus (N-1, 1)) = ? \tag{28}$$

The tensor products with the singlet representation are trivial, as it acts as the identity element, i.e. $(N) \otimes \lambda = \lambda$. Therefore, in this case we only need to determine the square of the standard representation. More generally, it is enough to compute tensor products of the form

$(N-1,1) \otimes \lambda$, which obey a simple rule. The result contains all representations obtained by moving the position of one box in the young tableau $\lambda$, with multiplicity one along with $\lambda$ itself with a multiplicity given the number of different row lengths of $\lambda$ minus one. For the two-index case at hand, this means

$$(N-1,1) \otimes (N-1,1) = (N) \oplus (N-1,1) \oplus (N-2,2) \oplus (N-2,1,1), \tag{29}$$

where we introduced the symmetric two-index representation $(N-2,2)$ of dimension $N(N-3)/2$ and the anti-symmetric two-index representation $(N-2,1,1)$ of dimension $(N-1)(N-2)/2$. To compute higher-fold tensor products, we simply take $(N-1,1)$ and continue performing the tensor product with the resulting representations. For example,

$$(N-1,1) \otimes (N-2,2) = (N-1,1) \oplus (N-2,2) \oplus (N-2,1,1) \oplus (N-3,3) \oplus (N-3,2,1), \tag{30}$$

where we now have a symmetric three-index tensor $(N-3,3)$ and a hook-like tensor $(N-3,2,1)$. Since $L_{-1}^i |0\rangle = 0$ (and any occurrence of $L_{-1}^i$ can be moved to the right with the Virasoro commutation relations), each additional $O(N)$ index will increase the spin of a current by at least 2. Hence, to reach the spin-10 currents studied by [1], we need to consider up to 5-fold tensor products of the standard representation. For the divergences we can go up to an 8-fold tensor product.

## 3.2 Counting currents and divergences

To understand whether all currents lift in the infrared fixed point, we need to systematically account for all of them, as well as a sufficient number of their associated divergence candidates. To this end, it is useful to study the partition function in the UV where the theory factorizes. We have

$$Z_{\text{CMM, UV}}(\tau, \bar{\tau}) = Z_{m\to\infty}(\tau, \bar{\tau})^N, \tag{31}$$

where $Z_{m\to\infty}$ denotes the partition function of the $m$'th minimal model in the large $m$ limit. In this limit the Virasoro characters

$$\chi_h^{(c)}(\tau) = \text{Tr}_{\mathcal{V}_{c,h}}(q^{L_0 - \frac{c}{24}}), \qquad q \equiv \exp(2\pi i \tau), \tag{32}$$

simplify dramatically, as the nested null module structure of minimal model characters becomes dominated by the leading null states. We find

$$\chi_{(1,1)}^{(c\to1)}(\tau) = \frac{q^{-\frac{1}{24}}}{\phi(q)}(1-q), \qquad \chi_{(r,s)}^{(c\to1)}(\tau) = \frac{q^{h_{r,s}-\frac{1}{24}}}{\phi(q)}(1-q^{rs}), \tag{33}$$

such that the partition function of a single diagonal minimal model approaches

$$Z_{m\to\infty}(\tau, \bar{\tau}) = \sum_{1 \le r \le s}^{\infty} \left| \chi_{(r,s)}^{(c\to1)}(\tau) \right|^2. \tag{34}$$

On the other hand, as a $c = N$ theory, the decoupled models also admit an expansion of the partition function in terms of the corresponding characters:

$$\begin{aligned}
Z_{\text{CMM, UV}}(\tau, \bar{\tau}) &= \sum_{h,\bar{h}} d_{h,\bar{h}} \, \chi_h^{(c=N)}(\tau) \chi_{\bar{h}}^{(c=N)}(\bar{\tau}) \\
&\supset \sum_{n=0}^{\infty} d_{n,0} \chi_n^{(c=N)}(\tau)(1 + O(\bar{q})) + d_{n,1}\chi_n^{(c=N)}(\tau)(\bar{q} + O(\bar{q}^2)),
\end{aligned} \tag{35}$$

Table 2: List of degeneracies for all diagonal-Virasoro primary currents in the tensor product theory. Note that $J = 10$ has been included here even though lifting for some irreps has only been checked up to $J = 9$.

| $J$ | $d_{J,0}$ | $d_{J,0}^{(N=4)}$ | $d_{J,0}^{(N=5)}$ | $d_{J,0}^{(N=6)}$ |
|---|---|---|---|---|
| 0 | 1 | 1 | 1 | 1 |
| 1 | 0 | 0 | 0 | 0 |
| 2 | $N-1$ | 3 | 4 | 5 |
| 3 | 0 | 0 | 0 | 0 |
| 4 | $N(N-1)/2$ | 6 | 10 | 15 |
| 5 | $(N-1)(N-2)/2$ | 3 | 6 | 10 |
| 6 | $(N+4)N(N-1)/6$ | 16 | 30 | 50 |
| 7 | $(3+2N)(N-1)(N-2)/6$ | 11 | 26 | 50 |
| 8 | $(N(15+N)+2)N(N-1)/24$ | 39 | 85 | 160 |
| 9 | $(N(13+3N)-26)N(N-1)/24$ | 37 | 95 | 200 |
| 10 | $(N(N(36+3N)+81)-74)N(N-1)/120$ | 89 | 226 | 481 |

where in the last line we highlighted the contributions of operators corresponding to the additional conserved currents, with weights $h = n, \bar{h} = 0$, and the corresponding divergences of weights $h = n, \bar{h} = 1$. We note that now the $c = N$ characters read

$$\chi_h^{(N)}(\tau) = \frac{q^{h-\frac{N}{24}}}{\phi(q)}, \qquad \phi(q) = \prod_{k=1}^{\infty}(1-q^k), \tag{36}$$

since no null states exist in non-vacuum Virasoro representations with $c > 1$. Matching the left- and right-hand sides of equation (31), we find

$$\sum_{n=0}^{\infty} d_{n,0} \chi_n^{(c=N)}(\tau) = \left(\chi_{(1,1)}^{(c\to 1)}(\tau)\right)^N,$$
$$\sum_{n=0}^{\infty} d'_{n,1} \chi_n^{(c=N)}(\tau) = \binom{N}{4}\left(\chi_{(1,1)}^{(c\to 1)}(\tau)\right)^{N-4}\left(\chi_{(1,2)}^{(c\to 1)}(\tau)\right)^4, \tag{37}$$

where the prime in the second sum denotes that we selected only operators that contain four $\phi_{(1,2)}$ constituents, since only these are relevant for the subsequent computations of anomalous dimensions. Expanding both sides at small $q$ allows us to determine the degeneracies to an arbitrarily high order, but we were not able to guess a closed form expression. We therefore list the degeneracies up to spin 10 for arbitrary $N$ in Table 2. These degeneracies turn out to be a polynomial in $N$ of degree $\lfloor J/2 \rfloor$.

Clearly, the multiplicities must be expressible as a non-negative integer combination of dimensions of representations of $S_N$. It is of course very important to understand the refinement of the spectrum under global symmetry irreps. While the next subsection will introduce a refined partition function which achieves this decomposition, it is already possible to compute the multiplicities of irreps for some low spins in Table 2. For example, for $J = 4$ and $N$ generic we know from the explicit construction in [1] that there is one singlet. Writing an Ansatz in terms of representations whose dimensions are a polynomial of at most degree 2 in $N$,

$$\frac{N(N-1)}{2} = d_{4,0}^{(N-2,2)}\dim(N-2,2) + d_{4,0}^{(N-2,1,1)}\dim(N-2,1,1) + d_{4,0}^{(N-1,1)}\dim(N-1,1)+1, \tag{38}$$

has the unique integer solution $d_{4,0}^{(N-1,1)} = d_{4,0}^{(N-2,2)} = 1, d_{4,0}^{(N-2,1,1)} = 0$. Similarly, it is easy to work out that for $J = 5$ we have a single representation $d_{5,0}^{(N-2,1,1)} = 1$ and that for $J = 6$ we

Table 3: List of degeneracies for all diagonal-Virasoro primary divergences built out of four $\phi_{(1,2)}$ constituents in the tensor product theory.

| $J$ | $d'_{J+1,1}$ | $d'^{(N=4)}_{J+1,1}$ | $d'^{(N=5)}_{J+1,1}$ | $d'^{(N=6)}_{J+1,1}$ |
|---|---|---|---|---|
| 0 | $\binom{N}{4}$ | 1 | 5 | 15 |
| 1 | $3\binom{N}{4}$ | 3 | 15 | 45 |
| 2 | $\binom{N}{4}(N+1)$ | 5 | 30 | 105 |
| 3 | $\binom{N}{4}(4N-6)$ | 10 | 70 | 270 |
| 4 | $\binom{N}{4}(N(N+11)-18)/2$ | 21 | 155 | 630 |
| 5 | $\binom{N}{4}(N(5N+1)-8)/2$ | 38 | 305 | 1335 |
| 6 | $\binom{N}{4}(N(N+34)-18)(N-1)/6$ | 67 | 590 | 2775 |
| 7 | $\binom{N}{4}(N(N(2N+13)-29)+14)/2$ | 117 | 1110 | 5550 |
| 8 | $\binom{N}{4}(N(N(N+71)+142)-192)(N-1)/24$ | 197 | 2015 | 10725 |
| 9 | $\binom{N}{4}(N(N(7N+137)+34)-168)(N-1)/24$ | 326 | 3585 | 20250 |

have $d^{(N-3,3)}_{6,0}=1$, $d^{(N-2,2)}_{6,0}=2$, $d^{(N-2,1,1)}_{6,0}=1$, $d^{(N-1,1)}_{6,0}=3$ along with 2 singlets.[14] For $J \geq 7$, the unrefined partition function is no longer enough. However, already at spin 6 we notice an interesting phenomenon for low $N$. The representation $(N-3,3)$ only exists for $N \geq 6$ and its dimension

$$\dim(N-3,3) = \frac{N(N-1)(N-5)}{6}, \tag{39}$$

vanishes for $N = 5$ and is negative for $N = 4$. In fact this phenomenon becomes more and more frequent as we increase spin and start accessing representations of higher rank. Remarkably, at $J = 6$ we find that the multiplicities for $N = 5$ remain unchanged, while for $N = 4$ the multiplicity $d^{(2,2)}_{6,0} = 1$ is reduced by one. It would be interesting to understand whether identities of the type

$$\dim(N-3,3)|_{N=4} = -\dim(N-2,2)|_{N=4}, \tag{40}$$

can be properly understood in the context of the analytic continuation of $S_N$ representations [74], the so-called Deligne category $\widetilde{\mathrm{Rep}}\,S_N$ allowing predictions for low $N$ degeneracies without having to check them in a case by case basis as we did in this work.

We can similarly count the multiplicities of divergences which we present in Table 3. These grow much faster and are polynomials in $N$ of degree $\lfloor J/2 \rfloor + 4$, which is already quite suggestive that currents have a very high chance of lifting, especially as $N$ increases. However, not only must the number of divergences match or exceed the number of currents, it must do so for each individual irreducible representation of $S_N$. We will soon see that this is the case for all spins, irreducible representations and values $N$ except for a single case. For $J = 3$ and $N = 4$, we find that the multiplicities are $d'^{(4)}_{4,1} = 1$, $d'^{(3,1)}_{4,1} = 2$, $d'^{(2,1,1)}_{4,1} = 1$, accounting for the ten states. However, there are no divergences in the $(2,2)$ representation that are candidates to recombine with the corresponding spin 4 current! For the pure $\sigma$ deformation, this is a non-perturbative constraint ensuring the presence of additional currents. In fact, this is simply a global symmetry refined version of Zamolodchikov's famous counting argument for the integrability of deformed minimal models [26,75].[15] However, for our infrared CFTs, both $g_\sigma$ and $g_\epsilon$ are non-vanishing meaning that operators built on top of $\phi_{(1,3)}$ might recombine

---

[14]To argue this for $J = 6$, we also need to know that $d^{(N-3,1,1)}_{6,0} = 0$. This is clear because we can only have three indices for a spin-6 current if all of them are on $L_{-2}$ charges, the anti-symmetrization of which will give zero.

[15]The possibility of $\phi_{(1,3)}$-type operators recombining with the current along the RG flow of the pure $\sigma$ deformation is forbidden at finite $m$ by dimensional analysis. This complements the approach of [76] which established integrability via a Toda field theory construction. See also section 5.2.

with the current. In ordinary conformal perturbation theory this would manifest itself at third order, with one $\epsilon$ and two $\sigma$ insertions which should generically be non-vanishing. There is therefore no reason to expect that the current should remain conserved at the fixed point, even for this special case.

## 3.3 Refined partition functions

While in practice we will have to explicitly construct the currents and candidate divergences in each representation of $S_N$ to check if anomalous dimensions are non-zero, it is still valuable to perform the counting without having to construct them. To do this we define a partition function refined by the irreducible representation under which the operators transform

$$Z_\lambda(\tau, \bar{\tau}) = \text{Tr}_{\mathcal{H}_\lambda} q^{L_0 - \frac{c}{24}} \bar{q}^{\bar{L}_0 - \frac{c}{24}} = \sum_{h, \bar{h}} d^\lambda_{h, \bar{h}} q^{h - \frac{c}{24}} \bar{q}^{\bar{h} - \frac{c}{24}}, \tag{41}$$

where $\mathcal{H}_\lambda$ is the Hilbert space of operators transforming in the irrep $\lambda$[16] and $d^\lambda_{h, \bar{h}}$ are the sought-after degeneracies of operators with weights $(h, \bar{h})$ in irrep $\lambda$. We claim that the refined partition function can be computed through the following formula

$$Z_\lambda(\tau, \bar{\tau}) = \frac{1}{|S_N|} \sum_g \chi_\lambda(g) Z(\tau, \bar{\tau}; g) = \frac{1}{|S_N|} \sum_{[g]} |[g]| \chi_\lambda([g]) Z(\tau, \bar{\tau}; [g]), \tag{42}$$

where $|S_N| = N!$ is the order of the symmetric group $[g]$ denotes the conjugacy class of the group element $g$ and $|[g]|$ is the size of this class. Additionally we introduced the twisted partition function

$$Z(\tau, \bar{\tau}; g) \equiv \text{Tr}_{\mathcal{H}}(q^{L_0 - \frac{c}{24}} \bar{q}^{\bar{L}_0 - \frac{c}{24}} g) = \sum_{h, \bar{h}, \lambda} d^\lambda_{h, \bar{h}} q^{h - \frac{c}{24}} \bar{q}^{\bar{h} - \frac{c}{24}} \chi_\lambda(g), \tag{43}$$

where $\chi_\lambda(g)$ are the $S_N$ characters of representation $\lambda$. The last equality follows from the definition of $\chi$ as a trace and the fact that the global symmetry commutes with the conformal algebra, by definition. The twisted partition functions are easy to compute directly in terms of their untwisted counterpart since the group element $g$ acts by permuting the copies. Indeed, for a conjugacy class $[g]$ labeled by a partition of $N$ encoded in the row lengths $\lambda_i$ of the associated Young tableau with $n$ rows we find[17]

$$Z(\tau, \bar{\tau}; g) = Z(\tau, \bar{\tau}; [g]) = \prod_{i=1}^n Z(\lambda_i \tau, \lambda_i \bar{\tau}). \tag{44}$$

For example, for a permutation $(12)(3)(4)$, labeled by the partition $(2, 1, 1)$ we have

$$Z(\tau, \bar{\tau}; [(12)(3)(4)]) = Z(2\tau, 2\bar{\tau}) Z(\tau, \bar{\tau})^2. \tag{45}$$

To show why (44) holds, we make explicit use of the factorized Hilbert space. Using $|i_1, \ldots, i_N\rangle$ as a shorthand for $\left| h_{i_1}, \bar{h}_{i_1}, \ldots, h_{i_N}, \bar{h}_{i_N} \right\rangle$, we have

$$\begin{aligned} Z(\tau, \bar{\tau}; g) &= \sum_{i_1, \ldots, i_N} \langle i_1, \ldots, i_N | q^{L_0^1 + \cdots + L_0^N - \frac{c}{24}} \bar{q}^{\bar{L}_0^1 + \cdots + \bar{L}_0^N - \frac{c}{24}} | g \cdot i_1, \ldots, g \cdot i_N \rangle \\ &= \sum_{i_1, \ldots, i_N} q^{h_{i_1} + \cdots + h_{i_N} - \frac{c}{24}} \bar{q}^{\bar{h}_{i_1} + \cdots + \bar{h}_{i_1} - \frac{c}{24}} \delta_{i_1, g \cdot i_1} \cdots \delta_{i_N, g \cdot i_N} \\ &= \sum_{i_1, \ldots, i_n} q^{\lambda_1(h_{i_1} - \frac{c}{24})} \bar{q}^{\lambda_1(\bar{h}_{i_1} - \frac{c}{24})} \cdots q^{\lambda_n(h_{i_n} - \frac{c}{24})} \bar{q}^{\lambda_n(\bar{h}_{i_n} - \frac{c}{24})} = \prod_{i=1}^n Z(\lambda_i \tau, \lambda_i \bar{\tau}). \end{aligned} \tag{46}$$

---

[16]This is well defined since the global symmetry commutes with the stress-tensor and the dilatation operator is therefore block diagonal with each block associated to an irrep $\lambda$.

[17]See also [8] for a derivation of the twisted partition functions in the case of the $q$-state Potts model, where the $S_q$ symmetry acts on the single-site Hilbert space on a lattice.

Table 4: Character table for $S_4$ with the conjugacy classes labeled by a representative. The identity element is denoted by empty parentheses.

| $S_4$ | $[()]$ | $[(12)]$ | $[(12)(34)]$ | $[(123)]$ | $[(1234)]$ |
|---|---|---|---|---|---|
| Class Size | 1 | 6 | 3 | 8 | 6 |
| $\chi_{(4)}$ | 1 | 1 | 1 | 1 | 1 |
| $\chi_{(3,1)}$ | 3 | 1 | -1 | 0 | -1 |
| $\chi_{(2,2)}$ | 2 | 0 | 2 | -1 | 0 |
| $\chi_{(2,1,1)}$ | 3 | -1 | -1 | 0 | 1 |
| $\chi_{(1,1,1,1)}$ | 1 | -1 | 1 | 1 | -1 |

It is now straightforward to derive (42) using the character orthogonality formula

$$\frac{1}{|S_N|} \sum_{[g]} |[g]| \chi_\lambda([g]) \chi_{\lambda'}([g]) = \delta_{\lambda,\lambda'} . \tag{47}$$

Indeed, using the form (43) for the twisted partition function, we can multiply by a character $\chi_{\lambda'}(g)$ and sum over all group elements. Then, using the orthogonality relation (47), we immediately find the explicit form for the refined partition function (41), proving the master formula (42).

As an application let us write down the refined partition functions and degeneracies for the case $N = 4$. Degeneracies for higher values of $N$ are listed in Appendix A. First, we recall the character table of $S_4$ in Table 4. Then, we make use of the master formula (42) and of (44) to derive the refined partition functions for the singlet (4) and symmetric (2,2) representations

$$Z_{(4)}(\tau, \bar{\tau}) = \frac{1}{4!} \left[ Z(\tau, \bar{\tau})^4 + 6Z(2\tau, 2\bar{\tau})Z(\tau, \bar{\tau})^2 + 3Z(2\tau, 2\bar{\tau})^2 + 8Z(3\tau, 3\bar{\tau})Z(\tau, \bar{\tau}) + 6Z(4\tau, 4\bar{\tau}) \right],$$

$$Z_{(2,2)}(\tau, \bar{\tau}) = \frac{1}{4!} \left[ 2Z(\tau, \bar{\tau})^4 + 6Z(2\tau, 2\bar{\tau})^2 - 8Z(3\tau, 3\bar{\tau})Z(\tau, \bar{\tau}) \right], \tag{48}$$

with similar formulae for the remaining irreps. It is then straightforward to expand these into characters, obtaining the degeneracies in Table 5 for currents and Table 6 for divergence candidates below.

Table 5: List of degeneracies for all $N = 4$ diagonal-Virasoro primary currents distinguished by irreducible representations in the tensor product theory. We count the number of irreps using the same notation as (38). The number of states is obtained by multiplying by appropriate representation dimensions.

| $J$ | $d_{J,0}^{(4)}$ | $d_{J,0}^{(3,1)}$ | $d_{J,0}^{(2,2)}$ | $d_{J,0}^{(2,1,1)}$ | $d_{J,0}^{(1,1,1,1)}$ |
|---|---|---|---|---|---|
| 0 | 1 | 0 | 0 | 0 | 0 |
| 1 | 0 | 0 | 0 | 0 | 0 |
| 2 | 0 | 1 | 0 | 0 | 0 |
| 3 | 0 | 0 | 0 | 0 | 0 |
| 4 | 1 | 1 | 1 | 0 | 0 |
| 5 | 0 | 0 | 0 | 1 | 0 |
| 6 | 2 | 3 | 1 | 1 | 0 |
| 7 | 0 | 1 | 1 | 2 | 0 |
| 8 | 4 | 6 | 4 | 3 | 0 |
| 9 | 1 | 5 | 2 | 5 | 2 |
| 10 | 6 | 14 | 8 | 8 | 1 |

Table 6: List of degeneracies for all diagonal-Virasoro primary divergence candidates built from four $\phi_{(1,2)}$ operators distinguished by irreducible representations in the tensor product theory.

| $J$ | $d'^{(4)}_{J+1,1}$ | $d'^{(3,1)}_{J+1,1}$ | $d'^{(2,2)}_{J+1,1}$ | $d'^{(2,1,1)}_{J+1,1}$ | $d'^{(1,1,1,1)}_{J+1,1}$ |
|---|---|---|---|---|---|
| 0 | 1 | 0 | 0 | 0 | 0 |
| 1 | 0 | 1 | 0 | 0 | 0 |
| 2 | 0 | 1 | 1 | 0 | 0 |
| 3 | 1 | 2 | 0 | 1 | 0 |
| 4 | 2 | 3 | 2 | 2 | 0 |
| 5 | 2 | 6 | 3 | 4 | 0 |
| 6 | 5 | 10 | 6 | 6 | 2 |
| 7 | 7 | 17 | 9 | 13 | 2 |
| 8 | 12 | 27 | 18 | 21 | 5 |
| 9 | 18 | 46 | 25 | 37 | 9 |

## 3.4 Comments on finite $m$

A reasonable worry about the analysis of [1] is whether the lifting of currents is expected to persist when the spins $J$ of the currents start scaling with some positive power of $m$. Indeed, conformal perturbation theory presumably breaks down when this happens. We claim that the physics at sufficiently large but finite $m$ should not be modified substantially when compared to the large-$m$ expansion. Indeed, the fundamental mechanism that ensures the lifting of currents is the overwhelming growth of divergence candidate with which they can recombine. As we showed in the subsection above, at fixed $J$ the degeneracies of divergences $d'_{J,1}$ when compared to the degeneracies of currents $d_{J,0}$ scale as

$$\frac{d'_{J+1,1}}{d_{J,0}} \sim N^4,\qquad(49)$$

at large $N$, and are substantially larger at fixed $N$.

When we take $m$ to be finite, the spectrum is modified by the appearance of additional null states. This reduces the number of divergence candidate but also of initial currents. A simple exercise in Virasoro characters and counting shows that this effect has negligible consequences on the ability for the currents to recombine with their divergence candidates. We recall that the for the $m$'th minimal model, the Virasoro character for a doubly-degenerate primary reads

$$\chi^{(c_m)}_{(r,s)}(\tau) = \frac{q^{-c_m/24}}{\phi(q)}\left(q^{h_{r,s}} + \sum_{k=1}^{\infty}(-1)^k(q^{h_{r+mk,(-1)^k s+(1-(-1)^k)(m+1)/2}} + q^{h_{r,k(m+1)+(-1)^k s+(1-(-1)^k)(m+1)/2}})\right).\qquad(50)$$

To illustrate what we just described let us specialize to the case of coupled tricritical Ising models, i.e. $m = 4$. In this case new null states for currents appear only at $J \geq 12$, while new null states for divergences appear at $J \geq 9$. However, the reduction of states is very mild. We list the degeneracies for $N = 5$ and $m \in \{4, \infty\}$ for both currents and divergences in Table 7. We see that the putative divergences continue to vastly outnumber the currents even for $J = m^2 = 16$. In fact, at large $N$ one observes that the reduction in the number of states is suppressed by at least $1/N^4$ compared to the overall number of states both for currents and divergences. This counting can be refined for each individual irreducible representation as detailed in section 3.3 and the mechanism that ensures lifting of the currents should therefore remain robust at large but finite $m$.

Table 7: List of degeneracies for diagonal-Virasoro primary currents and divergences built out of four $\phi_{(1,2)}$ for $N = 5$ and $m = 4$ as well as the corresponding values for $m = \infty$.

| $J$ | $d_{J,0}(m=4)$ | $d_{J,0}(m=\infty)$ | $d'_{J+1,1}(m=4)$ | $d'_{J+1,1}(m=\infty)$ |
|---|---|---|---|---|
| 9 | 95 | 95 | 3565 | 3585 |
| 10 | 226 | 226 | 6190 | 6250 |
| 11 | 294 | 294 | 10530 | 10670 |
| 12 | 588 | 593 | 17605 | 17950 |
| 13 | 815 | 815 | 28970 | 29750 |
| 14 | 1480 | 1500 | 46990 | 48600 |
| 15 | 2116 | 2136 | 75220 | 78460 |
| 16 | 3584 | 3654 | 118980 | 125255 |

## 4 Broken conservation and the fate of the infrared

We will now explain how to turn the above results into a concrete algorithm for strengthening the evidence that coupled minimal model fixed points are irrational. This leads to the main result of this paper which is that currents of spin strictly less than 10 outside the $\widehat{\mathfrak{Vir}}$ identity multiplet all lift for $N = 5, 6, 7$.[18] The steps below will be similar to those that were used (and explained very tersely) in [1] but without any assumptions about the currents or divergences being singlets of $S_N$.

### 4.1 Construction of operators

The previous section has given an algorithm to determine $d_{J,0}$ which is the dimension of the $(h, \bar{h}) = (J, 0)$ Verma module in the UV theory. Individual multiplicities of irreps $d_{J,0}^\lambda$, satisfying $\sum_\lambda d_{J,0}^\lambda \dim \lambda = d_{J,0}$, were also calculated. To go further, it is important to actually build the currents in this Verma module so that their fate in the infrared can be determined by the aforementioned recombination technique. We have done this in a two step process — first by generating a set of currents that form a basis when $N$ is arbitrarily large and second by filtering the resulting list for finite values of $N$. At large $N$, the number of linearly independent currents for each irreducible representation and spin are given in Table 8.

Let us now describe the first step in more detail. Each term in a singlet current is built from $a$ strings with $b_j$ Virasoro charges in the $j$'th one as

$$\sum_{i_1,\dots,i_a=1}^{N} \prod_{j=1}^{a} \prod_{k=1}^{b_j} L_{-n_{j,k}}^{i_j}, \tag{51}$$

where the sum runs over tuples $(i_1, \dots, i_a)$ of distinct elements.[19] In the notation of [1], this is written as

$$\left( \Sigma L_{-n_{1,1}} \dots L_{-n_{1,b_1}} \right) \dots \left( \Sigma L_{-n_{a,1}} \dots L_{-n_{a,b_a}} \right). \tag{52}$$

It is straightforward in principle to take a linear combination of all terms such that the total spin is $J$ and fix coefficients by demanding that the state is annihilated by $L_1^1 + \dots + L_1^N$ and

---

[18]Once $N$ is raised beyond a critical value (which depends on the spin and is 7 for spin 10), the number of UV currents does not change. It is therefore possible to run the $N = 5, 6, 7$ algorithm for general $N$ and make stronger conclusions. The obstacle here is simply that calculations are slower because coefficients that operators need to be primary are no longer numbers but rational functions of $N$.

[19]As a typographical note, the subscript of the Virasoro charge is $n_{j,k}$. We need both $j$ and $k$ because there are multiple copies and potentially multiple Virasoro charges being multiplied for each copy.

Table 8: The degeneracies of spin $J \leq 10$ currents which are relevant for the first step of our algorithm for constructing operators. They have been refined by $S_N$ representation under the assumption that $N \gg 1$. These can be predicted by following the finite $N$ techniques of section 3 by raising $N$ until the numbers stabilize. Note that the final analysis has been limited to $J \leq 9$ due to memory constraints.

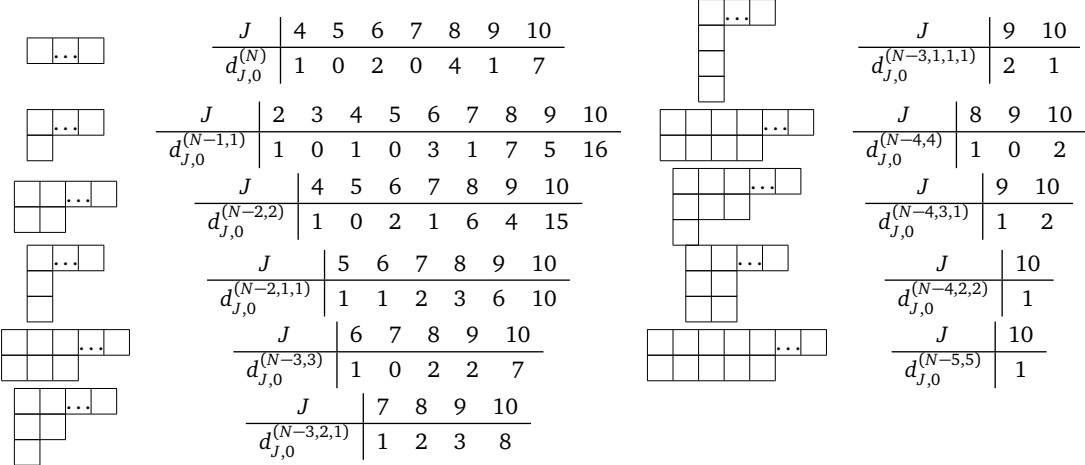

| $J$ | 4 | 5 | 6 | 7 | 8 | 9 | 10 |
|---|---|---|---|---|---|---|---|
| $d_{J,0}^{(N)}$ | 1 | 0 | 2 | 0 | 4 | 1 | 7 |

| $J$ | 2 | 3 | 4 | 5 | 6 | 7 | 8 | 9 | 10 |
|---|---|---|---|---|---|---|---|---|---|
| $d_{J,0}^{(N-1,1)}$ | 1 | 0 | 1 | 0 | 3 | 1 | 7 | 5 | 16 |

| $J$ | 4 | 5 | 6 | 7 | 8 | 9 | 10 |
|---|---|---|---|---|---|---|---|
| $d_{J,0}^{(N-2,2)}$ | 1 | 0 | 2 | 1 | 6 | 4 | 15 |

| $J$ | 5 | 6 | 7 | 8 | 9 | 10 |
|---|---|---|---|---|---|---|
| $d_{J,0}^{(N-2,1,1)}$ | 1 | 1 | 2 | 3 | 6 | 10 |

| $J$ | 6 | 7 | 8 | 9 | 10 |
|---|---|---|---|---|---|
| $d_{J,0}^{(N-3,3)}$ | 1 | 0 | 2 | 2 | 7 |

| $J$ | 7 | 8 | 9 | 10 |
|---|---|---|---|---|
| $d_{J,0}^{(N-3,2,1)}$ | 1 | 2 | 3 | 8 |

| $J$ | 9 | 10 |
|---|---|---|
| $d_{J,0}^{(N-3,1,1,1)}$ | 2 | 1 |

| $J$ | 8 | 9 | 10 |
|---|---|---|---|
| $d_{J,0}^{(N-4,4)}$ | 1 | 0 | 2 |

| $J$ | 9 | 10 |
|---|---|---|
| $d_{J,0}^{(N-4,3,1)}$ | 1 | 2 |

| $J$ | 10 |
|---|---|
| $d_{J,0}^{(N-4,2,2)}$ | 1 |

| $J$ | 10 |
|---|---|
| $d_{J,0}^{(N-5,5)}$ | 1 |

$L_2^1 + \cdots + L_2^N$. In the non-singlet case, we need to modify the above construction to allow for $c$ free indices as well:

$$\left( L_{-n_{1,1}}^{i_1} \ldots L_{-n_{1,b_1}}^{i_1} \right) \ldots \left( L_{-n_{c,1}}^{i_c} \ldots L_{-n_{c,b_c}}^{i_c} \right) ,$$
$$\left( \Sigma L_{-n_{c+1,1}} \ldots L_{-n_{c+1,b_{c+1}}} \right) \ldots \left( \Sigma L_{-n_{c+a,1}} \ldots L_{-n_{c+a,b_{c+a}}} \right) . \tag{53}$$

Again, there is no loss of generality in taking $(i_1, \ldots i_a)$ to all be different from each other. The simplest example is of course the multiplet of spin-2 currents in the standard representation corresponding to the individual Virasoro symmetries which break in the infrared. These can be expressed in the above notation as

$$L_{-2}^i - \frac{1}{N} \Sigma L_{-2} . \tag{54}$$

Notice that there are terms with both zero free indices and one free index. When we are interested in a representation with up to $c$ free indices, we can generate higher spin analogues of (54) in a loop from 0 to $c$. In a given iteration, we generate strings of charges which have the desired number of indices but do not necessarily have a total spin of $J$. We then make up the remaining units of spin by combining each such string with a singlet in all possible ways. Before applying $L_1^1 + \cdots + L_1^N$ and $L_2^1 + \cdots + L_2^N$ to this type of an Ansatz, one should also account for index symmetry. For the totally symmetric or totally anti-symmetric irreps, this is easy to do. For hook-like irreps, we need to choose whether to symmetrize or anti-symmetrize indices first since these operations do not commute. If we symmetrize first in the example of $\begin{array}{|c|c|} \hline i & j \\ \hline k \\ \cline{1-1} \end{array}$

$$T^{ijk} \mapsto T^{ijk} + T^{jik} - T^{kji} - T^{jki} , \tag{55}$$

which is what should be padded with singlets on the last ($c = 3$) iteration of the loop. For the earlier iterations, terms that still have hook symmetry with fewer indices are generated by deleting one of the positions in (55). Deleting the second and third lead to

$$T^{ik} + T^{jk} - T^{ki} - T^{ji} ,$$
$$T^{ij} + T^{ji} - T^{kj} - T^{jk} , \tag{56}$$

respectively. If we also delete the first, the terms this generates will be in the span of (56) due to symmetrization. Redundant terms can also appear in less trivial ways so we have chosen to (i) symmetrize first, (ii) delete positions in all inequivalent ways, (iii) pad these strings with singlets and (iv) check linear independence by explicit row reduction before adding such newly generated terms to the Ansatz. Enforcing primality of this Ansatz can be done in a runtime which is independent of $N$ due to the crucial assumption that none of $(i_1, \ldots, i_a)$ are the same. Indeed,

$$
\begin{aligned}
\sum_{i=1}^{N} L_1^i L_{-n_1}^j \ldots L_{-n_b}^j &= \left[ L_1^j, L_{-n_1}^j \ldots L_{-n_b}^j \right] + L_{-n_1}^j \ldots L_{-n_b}^j \sum_{i=1}^{N} L_1^i, \\
\sum_{i=1}^{N} L_1^i \Sigma L_{-n_1} \ldots L_{-n_b} &= \Sigma L_1 L_{-n_1} \ldots L_{-n_b} + (\Sigma L_{-n_1} \ldots L_{-n_b}) \Sigma L_1,
\end{aligned}
\tag{57}
$$

can be used repeatedly and the same goes for $L_2$. Examples of primaries we have produced in this way are

$$
\begin{aligned}
T_1 &= (\Sigma L_{-2})(\Sigma L_{-2}) - \frac{5}{9} \Sigma L_{-2} L_{-2} + \frac{1}{3} \Sigma L_{-4}, \\
T_2 &= L_{-2}^i L_{-2}^i - \frac{3}{5} L_{-4}^i - \frac{27}{5} L_{-2}^i \Sigma L_{-2} + \frac{1}{2} \Sigma L_{-2} L_{-2} - \frac{3}{10} \Sigma L_{-4}, \\
T_3 &= 2 L_{-2}^i L_{-2}^j - (L_{-2}^i + L_{-2}^j) \Sigma L_{-2} + \frac{5}{27} \Sigma L_{-2} L_{-2} - \frac{1}{9} \Sigma L_{-4},
\end{aligned}
\tag{58}
$$

for $N = 4$. An important comment at this stage is that nothing in the algorithm we have outlined so far guarantees that it will produce irreps. Indeed, the element of (58) which is in the $(2, 2)$ irrep is not $T_3$ but $T_3 + 3 T_1$. When constructing primaries that have $c$ free indices and a prescribed index symmetry, the resulting basis for the kernel of $L_1^1 + \cdots + L_1^N$ and $L_2^1 + \cdots + L_2^N$ generally mixes primaries in all irreps compatible with this symmetry where the number of free indices may be less than $c$. The second step of our procedure, to be discussed next, makes it possible to determine the linear combinations that correspond to irreps. Given a set of UV currents and divergence candidates which form a basis for finite $N$, one can proceed iteratively in $c$ by computing two-point functions and reaching irreps by demanding orthogonality. The main roadblock here is that going from an infinite $N$ basis to a finite $N$ basis only appears to be practical for the currents. For this reason, we have chosen to simply verify the lifting of currents without sorting them into irreps.[20]

The second step, which we will perform for currents but not divergences, is about making sure that the operators we have generated remain linearly independent when the finiteness of $N$ is taken into account. Two things to do in this regard are quite obvious. Some representations of $S_N$ expressed in terms of $N$ simply do not exist for $N$ small enough so currents containing these components never need to be generated in the first place. As an example, the $(N-3, 3)$ irrep first exists for $N = 6$ leading to the spin-6 primary

$$
\begin{aligned}
T_4 ={}& 6 L_{-2}^i L_{-2}^j L_{-2}^k - 3(L_{-2}^i L_{-2}^j + L_{-2}^i L_{-2}^k + L_{-2}^j L_{-2}^k) \Sigma L_{-2} \\
&+ (L_{-2}^i + L_{-2}^j + L_{-2}^k) \Sigma L_{-2} \Sigma L_{-2} - \frac{14}{15} \Sigma L_{-2} L_{-2} L_{-2} + \frac{5}{32} \Sigma L_{-3} \Sigma L_{-3} \\
&- \frac{343}{240} \Sigma L_{-3} L_{-3} - \frac{1}{4} \Sigma L_{-4} \Sigma L_{-2} + \frac{119}{30} \Sigma L_{-4} L_{-2} + \frac{97}{60} \Sigma L_{-6},
\end{aligned}
\tag{59}
$$

---

[20]An aesthetically pleasing (but also computationally intensive) alternative would be projecting currents in the kernel onto irreducible representations after the fact. The projectors would have to be formed from invariant tensors of $S_N$ which can be counted with the techniques of [77].

for $N = 6$ and no equivalent for $N = 4, 5$. It is also possible to save some effort by never generating terms which multiply $a > N$ strings of Virasoro charges. These vanish because of our assumption that copy indices are all different but this phenomenon cannot occur below spin 10. These tricks only get us so far and we must ultimately take a more brute force approach to discarding currents which are not linearly independent. As with the discussion under (56), this involves representing the currents as rows in a matrix and then reducing it. What makes this matrix especially large is that it cannot be formed from currents which are stored in a format independent of $N$. The data structures for the currents need to be unpacked so that the sums over $N$ are completely explicit. As such, the runtime of this procedure does depend on $N$ but we have been able to carry it out for currents up to spin 10 without much difficulty. The result is that we have to remove

$$J = 10 : \quad (4, 3), \tag{60}$$

for $N = 7$,

$$
\begin{aligned}
J &= 10 : \quad 2(3, 3) \oplus (4, 2), \\
J &= 8 : \quad (3, 3),
\end{aligned}
\tag{61}
$$

for $N = 6$,

$$
\begin{aligned}
J &= 10 : \quad 2(2, 2, 1) \oplus 2(3, 2) \oplus (4, 1), \\
J &= 9 : \quad (2, 2, 1), \\
J &= 8 : \quad (3, 2),
\end{aligned}
\tag{62}
$$

for $N = 5$ and

$$
\begin{aligned}
J &= 10 : \quad 2(2, 1, 1) \oplus 6(2, 2) \oplus (3, 1), \\
J &= 9 : \quad (2, 1, 1) \oplus 2(2, 2), \\
J &= 8 : \quad 2(2, 2) \oplus (3, 1), \\
J &= 6 : \quad (2, 2),
\end{aligned}
\tag{63}
$$

for $N = 4$, in keeping with the degeneracies in Appendix A. We can say more about the simplest example by considering the spin-6 currents with four copies. Those in the singlet representation (4), first constructed in [1], are

$$
\begin{aligned}
T_5 &= \frac{97}{50} \Sigma L_{-6} + \frac{119}{25} \Sigma L_{-4} L_{-2} - \frac{1}{2} \Sigma L_{-2} \Sigma L_{-4} - \frac{343}{200} \Sigma L_{-3} L_{-3} \\
&\quad + \frac{5}{16} \Sigma L_{-3} \Sigma L_{-3} - \frac{28}{25} \Sigma L_{-2} L_{-2} L_{-2} + \Sigma L_{-2} \Sigma L_{-2} \Sigma L_{-2}, \\
T_6 &= \frac{193}{50} \Sigma L_{-6} + \frac{236}{25} \Sigma L_{-4} L_{-2} - \frac{3}{2} \Sigma L_{-2} \Sigma L_{-4} - \frac{667}{200} \Sigma L_{-3} L_{-3} \\
&\quad + \frac{9}{16} \Sigma L_{-3} \Sigma L_{-3} - \frac{57}{25} \Sigma L_{-2} L_{-2} L_{-2} + \Sigma L_{-2} \Sigma L_{-2} L_{-2}.
\end{aligned}
\tag{64}
$$

Adding currents in the standard representation $(3, 1)$ to this list does not produce any over-completeness but one can instead add

$$
\begin{aligned}
T_7 &= 2 L_{-3}^i L_{-3}^j + \frac{16}{9} L_{-2}^i L_{-2}^j (L_{-2}^i + L_{-2}^j) - \frac{8}{3} (L_{-2}^i L_{-4}^j + L_{-4}^i L_{-2}^j) \\
&\quad + \frac{4}{5} \Sigma L_{-2} (L_{-4}^i + L_{-4}^j) - \Sigma L_{-3} (L_{-3}^i + L_{-3}^j) - \frac{96}{5} \Sigma L_{-2} \Sigma L_{-2} (L_{-2}^i + L_{-2}^j) \\
&\quad + \frac{16}{9} \Sigma L_{-2} L_{-2} (L_{-2}^i + L_{-2}^j) - \frac{4}{15} \Sigma L_{-4} (L_{-2}^i + L_{-2}^j) + \frac{9056}{1125} \Sigma L_{-2} L_{-2} L_{-2} - 2 \Sigma L_{-3} \Sigma L_{-3} \\
&\quad + \frac{42226}{3375} \Sigma L_{-3} L_{-3} + \frac{16}{5} \Sigma L_{-2} \Sigma L_{-4} - \frac{116464}{3375} \Sigma L_{-4} L_{-2} - \frac{47416}{3375} \Sigma L_{-6},
\end{aligned}
$$

$$T_8 = 2\Sigma L_{-2} L_{-2}^i L_{-2}^j - \Sigma L_{-2} \Sigma L_{-2}(L_{-2}^i + L_{-2}^j) + \frac{28}{75}\Sigma L_{-2} L_{-2} L_{-2} - \frac{5}{48}\Sigma L_{-3} \Sigma L_{-3}$$

$$+ \frac{343}{600}\Sigma L_{-3} L_{-3} + \frac{1}{6}\Sigma L_{-2}\Sigma L_{-4} - \frac{119}{75}\Sigma L_{-4} L_{-2} - \frac{97}{150}\Sigma L_{-6} \,, \tag{65}$$

which both appear to have a $(2,2)$ component in this notation. To check whether this is really true, we must appreciate the fact that the 8 types of terms written in (64) and the further 9 types of terms with free indices in (65) can only be regarded as independent abstract quantities when $N$ is sufficiently large. Since $N = 4$ here, we must represent these operators as vectors in a space of dimension 50 instead of 17. The basis is comprised of 4 $L_{-6}^i$ terms, 10 $L_{-3}^i L_{-3}^j$ terms, 16 $L_{-2}^i L_{-4}^j$ terms and 20 $L_{-2}^i L_{-2}^j L_{-2}^k$ terms. Once $T_5$ and $T_8$ are written in this higher dimensional format, it becomes easy to see that $T_8$ always comes out to be $-\frac{1}{3}T_5$ regardless of which values are chosen for its free indices $i$ and $j$. We therefore verify the last line of (63) by noticing that one operator designed to be new was secretly proportional to a singlet generated in a previous step.

Once operators in the $(h,\bar{h}) = (J,0)$ Verma module have been constructed using the two step process above, we need to repeat at least the first step for operators in $(h,\bar{h}) = (J+1,1)$ built using four copies of $\phi \equiv \phi_{(1,2)}$. This introduces a few ingredients which were not present for the currents. In particular, when we loop the number of free indices in a given term from 0 to $c$, there is a further loop over how many of these free indices should appear on strings that include a copy of $\phi$. Whether $\phi$ is part of a string with a free index or the singlet needed to achieve a spin of $J-1$, it should not be acted upon by $L_{-2}$ since it has a null state at level 2. As already anticipated in Table 3, the number of divergence candidates can easily be in the thousands due to the number of admissible terms in the Ansatz being similarly large. To apply $L_1^1 + \dots L_1^N$ and $L_2^1 + \dots L_2^N$ to this Ansatz more quickly, it is possible to only account for index symmetrizations (obtained from deleting positions in a master symmetrization) at the very end. Given a single term like $T^{ijk}$, one can apply $L_1^1 + \dots L_1^N$ and $L_2^1 + \dots L_2^N$ and then add the necesssary orbits under index permutations instead of doing this first. Even with this optimization however, our code for some irreps of interest ran out of memory at $J = 10$ thus forcing us to stop at $J = 9$. To remove redundant degrees of freedom from the space of divergence candidates, we have only implemented the trivial methods for this — focusing on irreps which actually exist at finite $N$ and omitting terms with $a > N$ strings. The explicit row reduction needed to find a basis for the divergences is very slow and fortunately not necessary. The extra operators in our collection just do one of two things.

1. For values of $N$ and $J$ which allow a current to stay conserved, these operators increase the number of three-point functions which need to be computed in order to verify this.

2. For values of $N$ and $J$ such that all currents lift, these operators *potentially* increase the number of three-point function computations although this depends on the order we choose for them. In practice this is only a slight slowdown.

In contrast to the divergences, removing redundant currents was crucial. If one of the operators in (60) through (63) were kept, there would be a linear combination of currents in our list which vanishes. A vanishing operator clearly has a vanishing anomalous dimension and therefore leads to a false positive in our search for currents that stay conserved.

## 4.2 Lifting from three-point functions

With explicit $(h,\bar{h}) = (J,0)$ primaries in hand, our goal is to show that none of them remain conserved in the IR for $N > 4$. In other words, the anomalous dimension matrix for each $J$ should have all non-zero eigenvalues at some loop order. This order is guaranteed not

to be $O(g_\sigma)$ because of chirality. It is also guaranteed not to be $O(g_\epsilon^n)$ for any $n$ since the pure $\epsilon$ deformation leaves minimal models decoupled. This leaves $O(g_\sigma^2)$ as the next simplest possibility which turns out to be the correct one. Once it is established that currents lift at this order, higher orders will not be able to restore conservation since we are working with asymptotically large $m$ and therefore small coupling.

The recipe of conformal perturbation theory tells us to show that there is no spin-$J$ current $T$ such that $\langle T(0)\sigma(z,\bar{z})\sigma(1)T(\infty)\rangle$ integrates to zero as a principal value. As advertised previously, conformal representation theory leveraged as in [39] shows that this can be replaced with a technically simpler check — we can equivalently show that there is no spin-$J$ current $T$ such that $\left\langle T(z_1)V^L(z_2,\bar{z}_2)\sigma(z_3,\bar{z}_3)\right\rangle$ vanishes for all spin-$(J-1)$ divergence candidates $V^L$. In practice, the algorithm described above produces $d_{J,0}$ operators $T^K$ and more than $d'_{J,1}$ operators $V^L$ because duplicates are slow to remove at finite $N$. From these data, we can proceed to compute the matrix

$$C_{L\sigma}^K = z_{12}^{2J-1} z_{13} z_{23} \bar{z}_{23}^2 \left\langle T^K(z_1) V^L(z_2,\bar{z}_2)\sigma(z_3,\bar{z}_3)\right\rangle, \tag{66}$$

and stop as soon as it becomes impossible to find a vanishing linear combination of the rows. In favourable cases, one can conclude this by computing only $d_{J,0}$ out of the more than $d'_{J,1}$ columns and seeing that this is already enough to give the matrix full rank.[21] Since the necessary matrix elements do not need to be computed in any particular order, it makes sense to write parallelized code for this purpose. Nevertheless, this task is not embarrassingly parallel because, even though the $N$ copy three-point functions are all different, they share a dependence on many of the same single-copy two-point and three-point functions. These building blocks should be memoized in a way that makes them accessible to each process.

We will now discuss the reduction to single-copy data. In $\left\langle T^K(z_1) V^L(z_2,\bar{z}_2)\sigma(z_3,\bar{z}_3)\right\rangle$, the first two operators are of course sums of several terms. For instance, (54) is a sum of 2 and (59) is a sum of 13. Looping over the terms in $T^K$ and the terms in $V^L$ will produce complicated three-point functions with each iteration but they will necessarily assemble into a three-point function of quasiprimaries at the end of the calculation. This will have the standard Polyakov form which guarantees that (66) is a pure number. A given term in $V^L$ will consist of a certain number of strings each of which is associated with a different copy. This number was called $a$ in the previous subsection. These strings should be paired up with strings from the $T^K$ term or strings from $\sigma \equiv \frac{1}{4!}\binom{N}{4}(\Sigma\phi)^4$ or both. This is what leads to the set of two-point and three-point functions mentioned above. We can account for all such pairings by taking $\sigma$ and the $T^K$ term to have a fixed ordering while permuting the $V^L$ term with respect to them. Table 9 shows some allowed and disallowed permutations for

$$
\begin{aligned}
T^{ij} &= L_{-2}^i L_{-2}^j \Sigma L_{-2}, \\
V^{ij} &= L_{-3}^i \phi^j (\Sigma L_{-1}\phi)(\Sigma L_{-1}\phi)(\Sigma\phi).
\end{aligned}
\tag{67}
$$

When these terms are taken from $T^K$ and $V^L$ operators that both have two indices, these operators actually give four chances to increase the rank of the matrix. The single term three-point function can have fully matching, partially matching or disjoint indices as in $\left\langle T^{ij}V^{ij}\sigma\right\rangle$, $\left\langle T^{ij}V^{ik}\sigma\right\rangle$, $\left\langle T^{ij}V^{kj}\sigma\right\rangle$ and $\left\langle T^{ij}V^{kl}\sigma\right\rangle$.

After we generate all of the permutations which are not eliminated by the conditions discussed in Table 9, they need to be weighted with the right combinatorial factors. For this, we simply count the number # of pairings where at least one of the copy indices is fixed (not part of a sum). The fourth case of Table 9 would have # = 3 because of the $(\Sigma L_{-2}, L_{-3}^k)$ and

---

[21]Clearly, it would be even more favourable to also sort the operators involved into $S_N$ irreps since these give (66) a block structure. We have had to skip this step because our methods are unable to ensure that all divergence candidates are linearly independent in a reasonable amount of time.

Table 9: A list of ways to permute the $V^{kj}$ strings in the $\left\langle T^{ij}V^{kj}\sigma\right\rangle$ three-point function. The first is disallowed because strings with $\phi$ are not matched. The second is disallowed because the strings with a free index of $j$ are not matched. The third is disallowed because strings with $i\neq k$ indices are matched. The fourth is allowed. Note that all would be disallowed if $V^{kj}$ were shorter than $T^{ij}$ or longer by more than 4 strings.

| $L^i_{-2}$ | $L^j_{-2}$ | $\Sigma L_{-2}$ | | | |
|---|---|---|---|---|---|
| $\Sigma\phi$ | $\phi^j$ | $L^k_{-3}$ | $\Sigma L_{-1}\phi$ | $\Sigma L_{-1}\phi$ | ✗ |
| | $\Sigma\phi$ | $\Sigma\phi$ | $\Sigma\phi$ | $\Sigma\phi$ | |

| $L^i_{-2}$ | | $L^j_{-2}$ | $\Sigma L_{-2}$ | | |
|---|---|---|---|---|---|
| $\Sigma L_{-1}\phi$ | $\Sigma L_{-1}\phi$ | $L^k_{-3}$ | $\Sigma\phi$ | $\phi^j$ | ✗ |
| $\Sigma\phi$ | $\Sigma\phi$ | | $\Sigma\phi$ | $\Sigma\phi$ | |

| $L^i_{-2}$ | $L^j_{-2}$ | $\Sigma L_{-2}$ | | | |
|---|---|---|---|---|---|
| $L^k_{-3}$ | $\phi^j$ | $\Sigma\phi$ | $\Sigma L_{-1}\phi$ | $\Sigma L_{-1}\phi$ | ✗ |
| | $\Sigma\phi$ | $\Sigma\phi$ | $\Sigma\phi$ | $\Sigma\phi$ | |

| $L^i_{-2}$ | $L^j_{-2}$ | $\Sigma L_{-2}$ | | | |
|---|---|---|---|---|---|
| $\Sigma\phi$ | $\phi^j$ | $L^k_{-3}$ | $\Sigma L_{-1}\phi$ | $\Sigma L_{-1}\phi$ | ✓ |
| $\Sigma\phi$ | $\Sigma\phi$ | | $\Sigma\phi$ | $\Sigma\phi$ | |

$(L^i_{-2},\Sigma\phi)$ and $(L^j_{-2},\phi^j)$ pairings. Going through all of the other pairings, the first comes with a factor of $N-\#$, the second comes with a factor of $N-\#-1$ and so on until we run out.

The algorithm above expresses a single term of $\left\langle T^K(z_1)V^L(z_2,\bar z_2)\sigma(z_3,\bar z_3)\right\rangle$ as a sum over permutations. We have seen that copy indices play a role in determining which permutations are absent from this sum and that the number of copies affects the coefficients for all remaining permutations. The rest of our task is insensitive to the index structure and the value of $N$ because it boils down to evaluating the terms of the sum which are products of single-copy correlators. These correlators come in three types.

1. Two-point functions where the primaries are $\phi,\phi$ and Virasoro descendants are only taken in the first position.

2. Three-point functions where the primaries are $\mathbf{1},\phi,\phi$ and there are Virasoro descendants in the first two positions.

3. Two-point functions with primaries $\mathbf{1},\mathbf{1}$ and descendants of both.

Correlators of the first type are by far the easiest. With all of the Virasoro charges hitting one operator, they can be pulled out one-by-one with

$$\left\langle L_{-n_1}\ldots L_{-n_b}\phi(z_2,\bar z_2)\phi(z_3,\bar z_3)\right\rangle = \mathcal{L}_{-n_1}\left\langle L_{-n_2}\ldots L_{-n_b}\phi(z_2,\bar z_2)\phi(z_3,\bar z_3)\right\rangle,$$
$$\mathcal{L}_{-n_1}\equiv \frac{n_1-1}{4(z_3-z_2)^{n_1}}-\frac{\partial_3}{(z_3-z_2)^{n_1-1}},\tag{68}$$

where we have used $h_\phi=\frac{1}{4}$. For correlators of the second type, it is only possible to use (68) if $\phi(z_2,\bar z_2)$ comes with either no Virasoro charges or simple powers of $L_{-1}$ which act as

derivatives. In general, this is not the case and we need to resort to a full evaluation of

$$\left\langle \prod_{i=1}^{b_1} L_{-n_{1,i}}(z_1) \prod_{j=1}^{b_2} L_{-n_{2,j}} \phi(z_2, \bar{z}_2) \phi(z_3, \bar{z}_3) \right\rangle \tag{69}$$

$$= \prod_{i=1}^{b_1} \oint_{z_1} \frac{dx_i}{2\pi i} (x_i - z_1)^{1-n_{1,i}} \prod_{j=1}^{b_2} \oint_{z_2} \frac{dy_j}{2\pi i} (y_j - z_2)^{1-n_{2,j}} \left\langle \prod_{k=1}^{b_1} T(x_k) \prod_{l=1}^{b_2} T(y_l) \phi(z_2, \bar{z}_2) \phi(z_3, \bar{z}_3) \right\rangle.$$

A slight optimization is to pull out whichever copies of $L_{-1}$ are present as derivatives first. The inner correlation function is computed iteratively with the Virasoro Ward identity and then the contour integrals are performed using the residue theorem. Even when this is done using specialized routines for rational functions, this is where the bulk of the computation time lies. For correlators of the third type, it is never possible to use (68) and we must return to (69) every time.

This concludes the list of operations that must be carried out in this approach to verify that UV currents lift. The arXiv submission for the latest version of this paper includes Python code for this purpose. It is sufficient to verify that anomalous dimensions are non-zero for the cases we have tested which are $J < 10$ and $N = 5, 6, 7$. It is worth emphasizing that these results are exact since the entries of (66) have been computed using rational numbers. With more resources, one could avoid having to stop at $N = 7$ (or indeed any fixed value) by working with rational functions of $N$.

## 5 Related constructions

### 5.1 More couplings: Adding fixed minimal models

An advantage of the construction in [1] is that it can be modified in a number of ways. Based on the mass lifting of both singlet and non-singlet currents exhibited here, it is reasonable to expect that further study along these lines will keep uncovering irrational CFTs with minimal chiral symmetry. Similar to what has been done for multiscalar CFTs close to the upper critical dimension [78–86], it is possible to consider couplings more general than (9) which fully or partially break $S_N$. We have previously mentioned orbifolds of coupled minimal models and these could again be considered for the subgroups of $S_N$ that are preserved. Additionally, similar constructions can be found which start from a tensor product of rational CFTs which each have $c > 1$. Here we would like to discuss something more conservative — a flow which preserves $S_N$ factors and stays within the domain of Virasoro minimal models.

The point is that for a product of operators from decoupled CFTs to become marginal from below as $m \to \infty$, not all of the individual scaling dimensions need to depend on $m$. For the $\phi_{(1,2)}$ operator in the $m$'th minimal model, which we will now write as $\phi_{(1,2)}(m)$, one option is combining this with three other copies of itself. But another is to combine it with $\phi_{(1,3)}(7)$ from the hexacritical Ising model. Indeed, since

$$h_{(1,2)}(m) = \frac{1}{4} - \frac{3}{4m} + O(m^{-2}), \qquad h_{(1,3)}(7) = \frac{3}{4}, \tag{70}$$

there is a new coupling which can be added when we take $N_1$ copies of the large $m$ minimal models and $N_2$ copies of the fixed one. Let us therefore amend the action (8) to be

$$S_{\text{new}} = \sum_{i=1}^{N_1} S_m^i + \sum_{i=1}^{N_2} S_7^i + \int d^2x \, g_\sigma \sigma + g_\epsilon \epsilon + g_\chi \chi, \tag{71}$$

with

$$\sigma \equiv \binom{N_1}{4}^{-\frac{1}{2}} \sum_{i<j<k<l}^{N_1} \phi^i_{(1,2)}(m)\phi^j_{(1,2)}(m)\phi^k_{(1,2)}(m)\phi^l_{(1,2)}(m),$$

$$\epsilon \equiv N_1^{-\frac{1}{2}} \sum_{i=1}^{N_1} \phi^i_{(1,3)}(m), \tag{72}$$

$$\chi \equiv (N_1 N_2)^{-\frac{1}{2}} \sum_{i=1}^{N_1}\sum_{j=1}^{N_2} \phi^i_{(1,2)}(m)\phi^j_{(1,3)}(7).$$

To see the new fixed points introduced by $\chi$, we do not need to know any OPE coefficients that are specific to the hexacritical Ising model. We can simply apply combinatorics to (12) once more to arrive at the beta-functions

$$\beta_\sigma = \frac{6}{m}g_\sigma - \frac{4\pi\sqrt{3}}{\sqrt{N_1}}g_\sigma g_\epsilon - 6\pi\binom{N_1-4}{2}\binom{N_1}{4}^{-\frac{1}{2}}g_\sigma^2,$$

$$\beta_\epsilon = \frac{4}{m}g_\epsilon - \frac{4\pi}{\sqrt{3N_1}}g_\epsilon^2 - \frac{2\pi\sqrt{3}}{\sqrt{N_1}}g_\sigma^2 - \frac{\pi\sqrt{3}}{2\sqrt{N_1}}g_\chi^2, \tag{73}$$

$$\beta_\chi = \frac{3}{2m}g_\chi - \frac{\pi\sqrt{3}}{\sqrt{N_1}}g_\chi g_\epsilon.$$

It appears that for most values of $N_1$, the only fixed points with $g_\chi^* \neq 0$ happen to have $g_\sigma^* = 0$. In particular, they have the expressions

$$g_{\epsilon\pm}^* = \frac{\sqrt{3N_1}}{2\pi m}, \qquad g_{\chi\pm}^* = \pm\frac{\sqrt{2N_1}}{\pi m}. \tag{74}$$

The only exceptions are $N_1 = 4$ with

$$g_{\epsilon\pm}^* = \frac{\sqrt{3}}{\pi m}, \qquad g_{\chi\pm}^* = \pm\frac{2}{\pi m}\sqrt{2 - \pi^2 m^2 g_\sigma^2}, \tag{75}$$

and $N_1 = 5$ with

$$g_{\epsilon\pm}^* = \frac{\sqrt{15}}{2\pi m}, \qquad g_{\chi\pm}^* = \pm\frac{\sqrt{2}}{\pi m}\sqrt{5 - 2\pi^2 m^2 g_\sigma^2}. \tag{76}$$

These apparent lines of fixed points without supersymmetry are most likely resolved at higher loops. For different examples of approximate conformal manifolds, see [87, 88]. Perhaps a more important point for us is that (74) can exist for $N_1 < 4$ because the $\sigma$ operator is not involved. For $(N_1, N_2) = (1, 1)$, $(N_1, N_2) = (1, 2)$ and $(N_1, N_2) = (2, 1)$, the fixed points for $m \to \infty$ have central charges of $\frac{53}{28}$, $\frac{39}{14}$ and $\frac{81}{28}$ respectively. All of these are smaller than the UV central charge of five tricritical Ising models which appear in the most optimistic way to get a CFT with low central charge and no enhanced symmetry by following [1]. The first one is also smaller than the central charge of three 3-state Potts models (1).

Continuing with this logic, it is not hard to see that we can also construct perturbative unitary flows by coupling $N_1$ large $m$ minimal models to $N_2$ ordinary critical Ising models. This time,

$$h_{(1,2)}(3) = \frac{1}{16}, \qquad h_{(1,3)}(3) = \frac{1}{2}, \tag{77}$$

are both important for building weakly relevant operators. The full system of beta-functions will involve couplings for

$$\mathcal{O}_1 \sim \sum_{i<j<k<l}^{N_1} \phi_{(1,2)}^i(m)\phi_{(1,2)}^j(m)\phi_{(1,2)}^k(m)\phi_{(1,2)}^l(m), \qquad \mathcal{O}_2 \sim \sum_{i=1}^{N_1} \phi_{(1,3)}^i(m),$$

$$\mathcal{O}_3 \sim \sum_{i<j}^{N_2} \phi_{(1,3)}^i(3)\phi_{(1,3)}^j(3), \qquad \mathcal{O}_4 \sim \sum_{i_1<\cdots<i_{16}}^{N_2} \phi_{(1,2)}^{i_1}(3)\ldots\phi_{(1,2)}^{i_{16}}(3),$$

$$\mathcal{O}_5 \sim \sum_{i_1<\cdots<i_{12}}^{N_2} \sum_{j=1}^{N_1} \phi_{(1,2)}^{i_1}(3)\ldots\phi_{(1,2)}^{i_{12}}(3)\phi_{(1,2)}^j(m),$$

$$\mathcal{O}_6 \sim \sum_{i_1<\cdots<i_8}^{N_2} \sum_{j<k}^{N_1} \phi_{(1,2)}^{i_1}(3)\ldots\phi_{(1,2)}^{i_8}(3)\phi_{(1,2)}^j(m)\phi_{(1,2)}^k(m), \qquad (78)$$

$$\mathcal{O}_7 \sim \sum_{i_1<\cdots<i_4}^{N_2} \sum_{j<k<l}^{N_1} \phi_{(1,2)}^{i_1}(3)\ldots\phi_{(1,2)}^{i_4}(3)\phi_{(1,2)}^j(m)\phi_{(1,2)}^k(m)\phi_{(1,2)}^l(m),$$

$$\mathcal{O}_8 \sim \sum_{i_1<\cdots<i_4}^{N_2} \sum_{j=1}^{N_2}\sum_{k=1}^{N_1} \phi_{(1,2)}^{i_1}(3)\ldots\phi_{(1,2)}^{i_4}(3)\phi_{(1,3)}^j(3)\phi_{(1,2)}^k(m),$$

$$\mathcal{O}_9 \sim \sum_{i=1}^{N_2}\sum_{j<k}^{N_1} \phi_{(1,3)}^i(3)\phi_{(1,2)}^j(m)\phi_{(1,2)}^k(m).$$

This makes it quite difficult to find the most general fixed points but it could also be useful to examine special cases. One of these involves the last operator $\mathcal{O}_9$ with $(N_1, N_2) = (2,1)$, which forms a closed OPE subsector along with $\mathcal{O}_2$. The UV for this theory has $c = \frac{5}{2}$ which is again smaller than the central charge of five tricritical Ising models. It is slightly larger than that of three 3-state Potts models but it may well be possible to make it smaller by taking $m$ below 11 in this construction and staying within the conformal window. An analysis of broken currents, as in this paper but for $S_{N_1} \times S_{N_2}$, will be important for learning more about these models.

## 5.2 Other probes of enhanced symmetry

Let us return to a discussion of the model (8) in the special case of $N = 4$. This is the one case we have found which has currents outside $\widehat{\mathfrak{Vir}}$ that stay conserved to two loops. For the representation (4), examples were constructed in [1] while this paper has additionally built conserved currents in (2, 2). Two points about this are worth emphasizing. First, the need for such operators (in some $S_N$ representation) is clear from integrability results. By mapping the $N = 4$ model onto an integrable Toda field theory, [76] showed indirectly that there must be enhanced symmetry when the pure $\sigma$ deformation is used to flow to a gapped phase. The currents that generate it must therefore survive the $O(g_\sigma^2)$ check we have done even if the $\epsilon$ deformation is able to lift them at a higher loop order. Second, the (2, 2) representation happens to lead to much stronger results than (4). As already mentioned, coupled minimal models do not yield any candidates for the divergences of such currents which means that they must be preserved by the $\sigma$ flow with either sign non-perturbatively.

There is another sense in which $N = 4$ preserves more symmetry than $N > 4$ and this comes from the notion of generalized symmetry [89]. Symmetry actions on local operators are implemented by topological operators of codimension 1 and in 2d these are topological defect lines [62]. If the fusion rules for topological defect lines take the form of a group law then they reproduce ordinary symmetries but more generally one must appeal to the concept

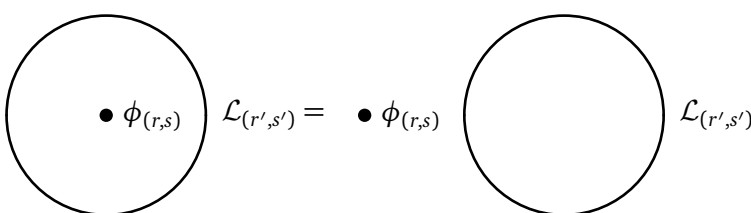

Figure 3: The condition for a non-invertible symmetry $\mathcal{L}_{(r',s')}$ to be preserved by a deforming operator $\phi_{(r,s)}$.

of a fusion category.[22] In particular, a topological defect line does not need to have an inverse. Minimal models have non-invertible symmetries implemented by Verlinde lines $\mathcal{L}_{(r,s)}$ which are labeled by the same integers as the primary operators and obey the same fusion rule.[23]

$$\mathcal{L}_{(r_1,s_1)} \times \mathcal{L}_{(r_2,s_2)} = \sum_{r_3=|r_1-r_2|+1}^{\min(r_1+r_2,2m-r_1-r_2)-1} \sum_{s_3=|s_1-s_2|+1}^{\min(s_1+s_2,2m+2-s_1-s_2)-1} \mathcal{L}_{(r_3,s_3)}. \tag{79}$$

Also in analogy with local operators, Verlinde lines associated to different decoupled copies in an $N$-fold tensor product fuse trivially into a line that we will call $\mathcal{L}^i_{(r_1,s_1)}\mathcal{L}^j_{(r_2,s_2)}$. In the following, we will show that certain types of these product lines are preserved along the flow only when $N = 4$.

Fortunately, there is a simple formula for how the line $\mathcal{L}_{(r',s')}$ acts on the primary $\phi_{(r,s)}$ in terms of the modular $S$-matrix

$$\mathcal{S}_{(r',s')(r,s)} = (-1)^{rs'+r's+1} \sqrt{\frac{8}{m(m+1)}} \sin\left(\pi r r' \frac{m+1}{m}\right) \sin\left(\pi s s' \frac{m}{m+1}\right). \tag{80}$$

The left side of Figure 3 picks up a factor of $\mathcal{S}_{(r',s')(r,s)}/\mathcal{S}_{(1,1)(r,s)}$ while the factor on the right side is the same but with $(1,1)$ in place of $(r,s)$. As reviewed in [90], we can set these equal to each other for $(r,s) = (1,3)$ to find the equation

$$\sin^2\left(\pi s' \frac{m}{m+1}\right) = \sin^2\left(\pi \frac{m}{m+1}\right). \tag{81}$$

To solve this within the standard Kac table, we need $s' = 1$ which restricts us to the subalgebra given by $\mathcal{L}_{(r',1)}$. Since $\epsilon$ in (8) is a sum of $\phi^i_{(1,3)}$ operators, it is clear that any product of lines which moves past $\epsilon$ trivially must be of the form $\prod_{i=1}^N \mathcal{L}^i_{(r_i,1)}$. A single other line in position $j$ would cause $\phi^j_{(1,3)}$ to pick up a non-trivial factor. Things are more interesting when we consider the $\sigma$ operator. Writing down the analogue of (81) for $(r,s) = (1,2)$ leads to

$$\cos\left(\pi s' \frac{m}{m+1}\right) = \cos\left(\pi \frac{m}{m+1}\right)(-1)^{r'-1}. \tag{82}$$

---

[22]The data of a fusion category can be used to identify the obstructions to gauging which were previously mentioned in section 2.3.

[23]To mention the simplest example, the critical Ising model ($m = 3$) has a single non-invertible Verlinde line and its action on local operators is the well known Kramers-Wannier duality.

In addition to the $s' = 1$ condition that we already derived, we further need $r'$ to be odd for each $\phi^i_{(1,2)}$ to commute with $\mathcal{L}_{(r',s')}$ individually. It is now easy to see that the $N = 4$ case allows $\prod_{i=1}^4 \mathcal{L}^i_{(r_i,1)}$ to include the even line $\mathcal{L}_{(2k,1)}$ an even number of times. As we commute this past the full product $\phi^1_{(1,2)}\phi^2_{(1,2)}\phi^3_{(1,2)}\phi^4_{(1,2)}$, signs from (82) will show up in pairs and therefore cancel. This is not true for $N > 4$. As long as $\phi^i_{(1,2)}\phi^j_{(1,2)}\phi^k_{(1,2)}\phi^l_{(1,2)}$ are allowed to have certain copies missing, there will only be two ways to prevent some such terms from picking up a sign. Namely acting on them with only $\mathcal{L}_{(2k+1,1)}$ lines or the product $\prod_{i=1}^N \mathcal{L}^i_{(r_i,1)}$ where all of the $r_i$ are even. A similar analysis can be done for the fixed minimal model cases of the previous subsection.

- For the model (71), it is easy to see that all preserved lines must be built out of $\mathcal{L}_{(2k+1,1)}$. This is because $\chi$ has a $\phi_{(1,2)}(m)$ operator appearing linearly.

- For an action which includes the operators $\mathcal{O}_2$ and $\mathcal{O}_9$ from (78), $\prod_{i=1}^{N_1} \mathcal{L}^i_{(r_i,1)}$ are again forced to have the $r_i$ be all even or all odd. This time, the minimal case of $N_1 = 2$ is not singled out as special.

The existence of such non-invertible lines in the presumably irrational CFTs in the IR of coupled minimal model flows was anticipated in [62] for the case of two-copy interactions and we see how this nicely generalizes to our four-copy case. It is also worth mentioning the work of [91] which establishes the existence of such lines for irrational supersymmetric CFTs that lie in the same conformal manifold as a rational CFT. Combined, these results give further credence to the expectation that non-invertible symmetries are ubiquitous in 2d CFTs and not just an artifact of rationality or exact solvability.

Another point we can make for general $N$ is that when $\sigma$ is used to flow to a TQFT in the IR, it is guaranteed to have degenerate vacua. This follows from the existence of topological defect lines which have a non-integer expectation value (empty circle) [62]. In such cases, [90] showed that the standard statement of crossing symmetry generically fails and needs to be replaced by a modified crossing equation which takes the normalizations of different vacua into account. A related phenomenon had previously been observed for Chern-Simons matter theories in [92]. It is therefore possible that integrable or non-integrable S-matrices which are reachable from coupled minimal models can be bootstrapped along the lines of [93].[24]

# 6 Outlook

While our understanding of compact irrational CFTs is very limited as it is, there is a further refinement of this space of theories that is even less understood and that is theoretically even more important: the space of interacting compact irrational CFTs with a *twist gap*. The twist gap is defined as the infimum of $\tau = \Delta - J$ over the spectrum of Virasoro primaries. In our work, we have given credible evidence that all operators in the IR CFT have positive twist, but we have not addressed how this twist behaves at large spin. While the general expectation based on arguments of convexity is that twist should *increase* with spin, it is a logical possibility that the twist actually decreases with spin such that the twist gap vanishes.[25] The existence of a positive twist gap is a key assumption of many modern universal results about 2d CFTs. The lightcone modular bootstrap [97–100] establishes the existence of families of operators whose weights approach $(c-1)/24$ and the Virasoro lightcone bootstrap [101, 102] further

---

[24]See also [94,95] for closely related work.

[25]In fact, for theories with a $u(1)$ chiral algebra it can be proven that the $u(1)$ twist gap vanishes [96].

establishes the existence of deformed double-twist trajectories.[26] Isolating the leading Regge trajectory in our models and understanding its shape will require intensive computations of full anomalous dimension matrices but will be important for answering whether or not we have identified CFTs that satisfy the assumptions of these key results. This could also allow us to test these predictions, guided by how a detailed numerical understanding of the 3d Ising model led to a verification of the power and accuracy of the analytic bootstrap in $d > 2$ [107].

Having slightly expanded the class of models we can study systematically in the $1/m$ expansion, it is also worth discussing just how far we might be able to push this logic. A straightforward step is to loosen the restriction of $S_N$ symmetry and allow for an arbitrary coupling to each set of copies. Concretely, we could consider the action

$$S = \sum_{i=1}^N S_m^i + \int d^2x \sum_{i<j<k<l}^N a_{ijkl} \phi_{(1,2)}^i \phi_{(1,2)}^j \phi_{(1,2)}^k \phi_{(1,2)}^l + \int d^2x \sum_{i=1}^N b_i \phi_{(1,3)}^i, \tag{83}$$

and do brute force searches for fixed points of the couplings $a_{ijkl}$ and $b_i$ in the spirit of [82] or attempt to constrain their general properties in the spirit of [79]. Additional parameters can be introduced by taking slightly different large $m$ limits for each copy, i.e we can take $m_i = m\gamma_i$ for different values of the constants $\gamma_i$. Furthermore, it is also possible to consider other classes of exactly solvable rational conformal field theories to be coupled in the UV. A straightforward example is to take the D-series minimal models, which also admit the $\sigma$ and $\epsilon$ deformations. Working at large but finite $m$, our analysis goes through since the same set of currents and divergence candidates is present. However, analytic continuation between the D-series minimal models appears to be more challenging. One therefore needs to keep in mind potential (but unlikely) cancellations between terms of different orders in the large-$m$ expansion. A richer generalization can be obtained by coupling minimal models of ADE type $\mathcal{W}$-algebras. In this case a classification of weakly relevant couplings is possible, but additional OPE coefficients must be determined to write down the beta-function equations. In this context, additional Virasoro primary currents are present because of the $\mathcal{W}$-symmetry. A relevant question is whether a diagonal copy of this symmetry can be preserved or whether only the Virasoro algebra survives. For example, it would be interesting to understand what the fate of diagonal $\mathcal{W}_3$ symmetry is in the coupled Potts models studied in [18].

Finally, to go beyond evidence and actually prove that the models in question are compact irrational CFTs with just Virasoro symmetry, we need to complement the brute force checks at finite spin by gaining some control over the large spin currents. One conceivable strategy is to bootstrap chiral algebras with a large spin gap between the stress-tensor and the first non-trivial generator, that is chiral algebras of the type $\mathcal{W}(2, J, J', \dots)$, for $J$ larger than the values we can explicitly check.[27] Some hope comes from considering what happens in the case of chiral algebras with a single non-trivial generator, $\mathcal{W}(2, J)$. While such algebras can exist for large values of $J$, they are only unitary for finitely many values of $c$, except when $J \leq 6$ [110]. In our case, if some enhanced chiral symmetry remains, it should do so at least for a small interval of $c$ below each integer $N$, where the chiral algebra is guaranteed to be unitary. Therefore, by studying the crossing equation for the four-point functions of the generators, and assuming unitarity in a range of values of $c$, it is conceivable that an upper bound on $J$ for $\mathcal{W}(2, J, J', \dots)$ algebras exists. Such a bound would allow us to stop our checks at finite spin and finally prove minimal chiral symmetry and hence irrationality.

---

[26]Along with trajectories of analytically continued scaling dimensions, a twist gap also influences the behaviour of averaged OPE coefficients involving heavy operators [103]. For boundary and crosscap analogues of this result, see [104–106].

[27]For previous examples of the numerical bootstrap being done with Virasoro blocks, see [108, 109].

# Acknowledgments

We are grateful for conversations with Christopher Beem, Nathan Benjamin, Jasper Jacobsen, Elias Kiritsis, Dalimil Mazáč, Hirosi Ooguri, Eric Perlmutter, Jiaxin Qiao, Sylvain Ribault, Junchen Rong, Slava Rychkov, Adar Sharon, Yifan Wang and Jingxiang Wu.

**Funding information**   AA received funding from the German Research Foundation DFG under Germany's Excellence Strategy - EXC 2121 Quantum Universe - 390833306 and is funded by the European Union (ERC, FUNBOOTS, project number 101043588). Views and opinions expressed are however those of the author(s) only and do not necessarily reflect those of the European Union or the European Research Council Executive Agency. Neither the European Union nor the granting authority can be held responsible for them. CB received funding from the São Paulo Research Foundation (FAPESP) grants 2019/24277-8, 2021/14335-0 and 2023/03825-2. AA thanks IPhT Saclay where part of this work was presented. CB thanks the Perimeter Institute for hospitality during the final stages of this work. Research at the Perimeter Institute is supported in part by the Government of Canada through NSERC and by the Province of Ontario through MRI.

# A   Further degeneracy tables

In this appendix we list irrep-refined degeneracies for $N = 5, 6$ using the methods of section 3.3.

Table 10: Degeneracies for all $N = 5$ $\widehat{\mathfrak{Vir}}$ primary currents.

| $J$ | $d_{J,0}^{(5)}$ | $d_{J,0}^{(4,1)}$ | $d_{J,0}^{(3,2)}$ | $d_{J,0}^{(3,1,1)}$ | $d_{J,0}^{(2,1,1,1)}$ | $d_{J,0}^{(2,2,1)}$ | $d_{J,0}^{(1,1,1,1,1)}$ |
|---|---|---|---|---|---|---|---|
| 0 | 1 | 0 | 0 | 0 | 0 | 0 | 0 |
| 1 | 0 | 0 | 0 | 0 | 0 | 0 | 0 |
| 2 | 0 | 1 | 0 | 0 | 0 | 0 | 0 |
| 3 | 0 | 0 | 0 | 0 | 0 | 0 | 0 |
| 4 | 1 | 1 | 1 | 0 | 0 | 0 | 0 |
| 5 | 0 | 0 | 0 | 1 | 0 | 0 | 0 |
| 6 | 2 | 3 | 2 | 1 | 0 | 0 | 0 |
| 7 | 0 | 1 | 1 | 2 | 0 | 1 | 0 |
| 8 | 4 | 7 | 5 | 3 | 0 | 2 | 0 |
| 9 | 1 | 5 | 4 | 6 | 2 | 2 | 0 |
| 10 | 7 | 15 | 13 | 10 | 1 | 6 | 0 |

Table 11: Degeneracies for all $N = 5$ $\widehat{\mathfrak{Vir}}$ primary divergence candidates.

| $J$ | $d'^{(5)}_{J+1,1}$ | $d'^{(4,1)}_{J+1,1}$ | $d'^{(3,2)}_{J+1,1}$ | $d'^{(3,1,1)}_{J+1,1}$ | $d'^{(2,1,1,1)}_{J+1,1}$ | $d'^{(2,2,1)}_{J+1,1}$ | $d'^{(1,1,1,1,1)}_{J+1,1}$ |
|---|---|---|---|---|---|---|---|
| 0 | 1 | 1 | 0 | 0 | 0 | 0 | 0 |
| 1 | 0 | 1 | 1 | 1 | 0 | 0 | 0 |
| 2 | 1 | 2 | 2 | 1 | 0 | 1 | 0 |
| 3 | 2 | 5 | 3 | 4 | 1 | 1 | 0 |
| 4 | 4 | 9 | 8 | 7 | 2 | 5 | 0 |
| 5 | 5 | 16 | 15 | 16 | 5 | 9 | 0 |
| 6 | 12 | 31 | 29 | 28 | 11 | 19 | 2 |
| 7 | 17 | 53 | 52 | 57 | 23 | 37 | 2 |
| 8 | 32 | 93 | 96 | 98 | 44 | 72 | 7 |
| 9 | 50 | 160 | 164 | 182 | 85 | 126 | 13 |

Table 12: List of degeneracies for all $N = 6$ $\widehat{\mathfrak{Vir}}$ primary currents. Additional irreps have zero degeneracy up to spin 10.

| $J$ | $d^{(6)}_{J,0}$ | $d^{(5,1)}_{J,0}$ | $d^{(4,2)}_{J,0}$ | $d^{(4,1,1)}_{J,0}$ | $d^{(3,1,1,1)}_{J,0}$ | $d^{(3,2,1)}_{J,0}$ | $d^{(3,3)}_{J,0}$ | $d^{(2,2,2)}_{J,0}$ |
|---|---|---|---|---|---|---|---|---|
| 0 | 1 | 0 | 0 | 0 | 0 | 0 | 0 | 0 |
| 1 | 0 | 0 | 0 | 0 | 0 | 0 | 0 | 0 |
| 2 | 0 | 1 | 0 | 0 | 0 | 0 | 0 | 0 |
| 3 | 0 | 0 | 0 | 0 | 0 | 0 | 0 | 0 |
| 4 | 1 | 1 | 1 | 0 | 0 | 0 | 0 | 0 |
| 5 | 0 | 0 | 0 | 1 | 0 | 0 | 0 | 0 |
| 6 | 2 | 3 | 2 | 1 | 0 | 0 | 1 | 0 |
| 7 | 0 | 1 | 1 | 2 | 0 | 1 | 0 | 0 |
| 8 | 4 | 7 | 6 | 3 | 0 | 2 | 1 | 0 |
| 9 | 1 | 5 | 4 | 6 | 2 | 3 | 2 | 0 |
| 10 | 7 | 16 | 14 | 10 | 1 | 8 | 5 | 1 |

Table 13: List of degeneracies for $N = 6$ $\widehat{\mathfrak{Vir}}$ primary divergence candidates built from four $\phi_{(1,2)}$ operators. Operators in irreps with no current to recombine with are not listed.

| $J$ | $d'^{(6)}_{J+1,1}$ | $d'^{(5,1)}_{J+1,1}$ | $d'^{(4,2)}_{J+1,1}$ | $d'^{(4,1,1)}_{J+1,1}$ | $d'^{(3,1,1,1)}_{J+1,1}$ | $d'^{(3,2,1)}_{J+1,1}$ | $d'^{(3,3)}_{J+1,1}$ | $d'^{(2,2,2)}_{J+1,1}$ |
|---|---|---|---|---|---|---|---|---|
| 0 | 1 | 1 | 1 | 0 | 0 | 0 | 0 | 0 |
| 1 | 0 | 1 | 1 | 1 | 0 | 1 | 1 | 0 |
| 2 | 1 | 3 | 3 | 2 | 0 | 2 | 1 | 1 |
| 3 | 2 | 6 | 6 | 6 | 2 | 5 | 3 | 0 |
| 4 | 5 | 12 | 15 | 11 | 4 | 13 | 6 | 3 |
| 5 | 6 | 22 | 27 | 26 | 11 | 28 | 13 | 5 |
| 6 | 15 | 45 | 57 | 49 | 24 | 58 | 25 | 14 |
| 7 | 22 | 78 | 103 | 100 | 54 | 119 | 50 | 25 |
| 8 | 44 | 145 | 199 | 183 | 106 | 231 | 92 | 57 |
| 9 | 69 | 255 | 352 | 349 | 215 | 438 | 173 | 102 |

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
