# Peer review of "Coupled minimal models revisited II: Constraints from permutation symmetry"

_SciPost Physics, doi:SciPost Phys. 18, 132 (2025)_

## Round 2 · Referee Report · Sylvain Ribault (Referee 1) · 2025-2-14

Report
Warnings issued while processing user-supplied markup:
- Inconsistency: plain/Markdown and reStructuredText syntaxes are mixed. Markdown will be used.
Add "#coerce:reST" or "#coerce:plain" as the first line of your text to force reStructuredText or no markup.
You may also contact the helpdesk if the formatting is incorrect and you are unable to edit your text.
Understanding the space of CFTs is a fundamental problem of mathematical physics, but CFTs are vastly more complicated than Lie algebras, and classifying all of them is far from achievable. However, one can start with smaller spaces of CFTs. First we can focus on 2d CFTs, which are more tractable thanks to their infinite-dimensional symmetry algebra. Then we can restrict to compact, unitary, modular-invariant CFTs, a physically well-motivated class which is expected to be sparse in the space of all CFTs. For simplicity we can further assume that there is a twist gap, which not only implies that there is no extended chiral symmetry, but also (together with modular invariance) leads to strong constraints on the spectrum.
The space of CFTs that obey all these assumptions is still expected to be vast, but there are no confirmed examples beyond minimal models. It is reasonably easy to construct candidate CFTs by perturbing products of minimal models and following the resulting RG flows, but it is much harder to prove that these candidates have a twist gap. The weaker statement that there is no extended chiral symmetry is very plausible in many cases, if only because extended symmetries typically do not arise by accident. Proving this statement remains technically challenging, and this is the challenge that the present article addresses.
So, we start from a product $\mathcal{P}_N$ of $N$ unitary diagonal minimal models. This product CFT has a large extended chiral symmetry: the product of the $N$ Virasoro algebras. It also has a permutation symmetry, described by the symmetric group $S_N$. Then we apply a perturbation that preserves permutation symmetry, and breaks chiral symmetry. But how do we know that no chiral symmetry is restored (or appears) at the fixed point, beyond conformal symmetry? The present article's approach is to focus on short RG flows, generated by nearly marginal operators. Such flows are accurately described by perturbation theory around the original product CFT $\mathcal{P}_N$. Unitary minimal models come with an integer parameter $m$, and they can be described perturbatively in $\frac{1}{m}$ near $m=\infty$: we therefore consider perturbing operators of dimension $1-O(\frac{1}{m})$, leading to fixed points at critical couplings $O(\frac{1}{m})$.
Consider chiral currents in the product CFT $\mathcal{P}_N$, i.e. primary fields $T^K$ of left and right conformal dimensions $(\Delta,\bar\Delta)=(J,0)$, which obey $\bar\partial T^K=0$. The breaking of the corresponding extended symmetries occurs if $\bar\partial T^K$ becomes nonzero along the flow. As explained in Sections 2.2 and 4.2, this is equivalent to the non-vanishing in $\mathcal{P}_N$ of 3-point functions of the type $\left<T^KV^L\sigma\right>$, where $\sigma$ is the perturbing (nearly marginal) operator and $V^L$ are primary fields of conformal dimensions $(\Delta,\bar\Delta)=(J,1)$. This non-vanishing of 3-point functions in a product of minimal models is now a tractable technical question, which the authors answer up to $J=10$ using computer calculations. Their result is that indeed, there are no extended symmetries.
How confident can we be that this result is true? This is hard to say, for two main reasons. First, we do no know for which values of the parameters $m$ and $N$ the result is supposed to hold. We know that $m$ should be large but finite, which must mean larger than some $N$-dependent bound. (Does the bound also depends on $J$?) The calculations described in Section 4.2 seem $N$-dependent, since they are about a matrix whose size $d_{J,0}$ depends on $N$, but it is not stated which values of $N$ were investigated. Second, the result cannot in principle be checked because it heavily relies on computer code that is not publicly accessible. And since the result is a negative statement, it would be hard to check by other means.
Nevertheless, this article is valuable for identifying large families of CFTs that are most probably unitary, compact, without extended chiral symmetry, and for introducing methods that could allow us to tame these CFTs. These methods include the analysis of the action of $S_N$ in Section 3. Although this analysis seems to play no role in the main result, it uncovers fundamental structures in the spectrum and its dependence on $N$. Somewhat paradoxically, while this work aims to explore CFT way beyond the well-known minimal models, it reaffirms the crucial importance of exactly solved models, since the CFTs under consideration are built from minimal models.
Main issues:
-
What exactly is proved in this article, and what is conjectured? First there is the statement from Section 4.2 that symmetries are broken by the perturbation. This does not seem to depend much on $m$, as explained in Section 3.3. But for which values of $J,N$ has this statement been established? The statement that the currents "appear to lift" in Section 4 is too vague. Then there is the statement that RG flows are short enough that the fixed point has no extended symmetry. This requires $m\geq m_0$, what can we say on $m_0$? How does $m_0$ depend on $J,N$? Can $m_0$ be estimated?
-
In Sections 3.2 and 4.1, the counting of currents and divergences by $S_N$ irrep is done in a pedestrian way, by constructing them explicitly. Such explicit constructions are needed in Section 4.2. Nevertheless, it would be interesting to have more general, analytic results on the decomposition of the spectrum into $S_N$ irreps. States are counted using partition functions: there are also twisted partition functions, which account for the action of $S_N$, and decompose into characters of the Virasoro algebra and of $S_N$. This works for determining the spectrum of the $q$-state Potts model, see https://arxiv.org/abs/2208.14298 . Could this work here?
-
Starting from the title, the article gives the impression that $S_N$ symmetry is crucial. But then there is the apparent admission in footnote 14 page 21 that $S_N$ irreps are not used to derive the main result. It would be good to state earlier and more explicitly which role $S_N$ symmetry plays.
Some other questions, and suggestions for improving the text:
-
Typos: encoutners, Verma modula, memoized, eachother, vaccua (twice), vaccuum (twice).
-
In the introduction, the questions Q1, Q2 about spaces of CFTs could be complemented by a Q3 about CFTs with a twist gap, instead of postponing the subject to the outlook.
-
The notation $\frac{3}{f}-\frac12$ for the dimension of $\epsilon^i$ in Eq. (1.1) differs from the rest of the article, where the same number would be called $h_{(1,2)}$. How is Eq. (1.1) related to Figure 1? Do they have the same range of values of $q$? Does the perturbation become marginal for $q=2$ and $q=4$?
-
Is it for technical simplicity that only diagonal minimal models are used, or is there another reason for not using non-diagonal minimal models?
-
The integer $m$ is called the rank of a minimal model, is this standard terminology?
-
On page 3, the statement that irrationality follows from modular invariance and $c>1$ could be clarified.
-
The discussion in Section 2.1 about recovering minimal models from generalized minimal models is nice, but is it relevant? At the end we focus on $\phi_{(1,2)}$ and $\phi_{(1,3)}$, which obey $r_i, s_i\ll m$.
-
In Eq. (2.7), parentheses would be welcome.
-
After Eq. (2.12), a factor $\sqrt{N}$ seems to be missing in $g^*_\epsilon$ for the fully stable fixed point.
-
In the legend of Figure 2, $FP_\pm^*$ could be called by their names, rather than the "fixed points in the middle".
-
Page 8, the explanation why checking primaries is sufficient could be clarified.
-
In Section 2.2, it would be nice to carry out the reasoning until the end, and clearly state the outcome: that we only need to compute $\left<T^KV^L\sigma\right>$ in the unperturbed CFT. Instead, this statement is postponed to Eq. (4.14), leaving us with Eq. (2.17) which looks not so easy to evaluate. Moreover, it would be useful to clearly define divergences and divergence candidates.
-
In Section 2.3, the reasoning that ends by invoking Hecke operators would deserve a conclusion.
-
In Eq. (3.1) should there be $N$?
-
In Section 3.1, most of the material is standard and covered in the Wikipedia article on Representation theory of the symmetric group. Does this material need to be repeated in each article that uses $S_N$ representations?
-
When writing $S_N$ representations, it might be simpler to remove the first line of the Young diagrams, and to write $(1)$ instead of $(N-1,1)$.
-
At the very end of Section 3.2, it would be nice to conclude on the meaning of the single case where there are fewer divergences than currents. Are you conjecturing that recombination occurs, even though the sufficient (but not necessary) condition of having enough divergences in the right $S_N$ irrep is not obeyed?
-
Why is there Eq. (3.19)? What should we deduce from it?
-
In Table 5, it would be good to state more explicitly what $n$ is, and to demonstrate the agreement with Table 2 in an example, for example $J=4$.
-
On page 20 the explanation that starts with "To remove redundant degrees of freedom" is obscure. Such technical details could either be explained more carefully (with examples) or omitted altogether.
-
The expression "Making this parallel" is awkward, it should be said more explicitly that this is about code and not physics.
-
In Section 4.2, instead of the statement "this is the last step which is sensitive...", it would be good to state more precisely how the calculations are sensitive to $N$ and to the index structure.
As a matter of principle I do not make recommendations on whether to publish articles or not. Editorial decisions belong to editors. Since the computer system makes it necessary to have a recommendation, I picked the first one in the list.
Recommendation
Publish (surpasses expectations and criteria for this Journal; among top 10%)
Author: Antonio Antunes on 2025-03-06 [id 5269]
(in reply to Report 1 by Sylvain Ribault on 2025-02-14)
Dear Sylvain,
We uploaded a new version of our paper to the arxiv, with a few important additions (let me highlight here a nicer version of the python code Connor wrote is now available with the source code and we computed partition function refined by irreducible representation to make the counting of states more streamlined).
Below I add a detailed response to your comments in the report. Thanks again for all your useful insight and suggestions.
Regarding your main concerns, we now address them in order.
-
What is actually proved is that for $J<10$ and $N=5,6,7$ all currents have non-vanishing anomalous dimension to leading order in the large $m$ expansion (this is stated in the first paragraph of section 4 and we now also do so in the last paragraph of section 4). We conjecture it to be true for all $N>4$ and all $J$. The question of what value of $m_0$ is sufficiently large is a subtle one. If $m_0$ is defined such that anomalous dimensions for all $J$ are non-vanishing, this could have a potentially non-trivial $N$ dependence. Conformal perturbation theory is expected (at the level of lore) to have a finite radius of convergence, which would allow us to estimate $m_0$, but it is not known how to estimate this radius. We did not discuss this explicitly since almost nothing is proved at the technical level, but we can mention this expectation if it improves the clarity of the presentation. Nonetheless, we emphasize, that even for zero radius of convergence (as is the case in usual Feynman diagram perturbation theory), the predictions for anomalous dimensions would be reliable, since the error term is still bounded.
-
Indeed, it is fruitful to use twisted partition functions to determine degeneracies of operators in arbitrary representations before constructing them. We have expanded section 3 so that this approach is now given a proper treatment.
-
In some sense the role of S_N symmetry is to make our life harder, since we have more conditions to verify. While we are sometimes agnostic about which irrep (if any) a current belongs to, going beyond the singlet sector is crucial to establish our main result. Besides adding some clarification to the footnote you mention, we believe that the use of refined partition functions in the new version makes it clear that S_N symmetry is helpful for thinking about the problem.
In response to your other suggestions, we have the following thoughts.
-
Fixed.
-
While we agree that such a Q3 feels thematically consistent with the rest of the introduction, we find that our results do not present strong evidence in this direction, since we do not explicitly evaluate the anomalous dimensions. We have made clearer in the conclusions that this is work to be done in the future.
-
We have added a bit to the discussion under Eq. (1.1) and mentioned that the deformation is marginal for $q = 2$. It is not marginal for $q = 4$ which means that Figure 1 had an error. We had it confused with the figure that applies to coupled Ising models in dimension between 2 and 4. While we agree that one finds $h_{(1,2)}$ by comparing the dimension of $\epsilon^i$ to the central charge, we would prefer to wait before introducing the Kac table notation.
-
Much of our comfort level with this analytic continuation came from the fact that diagonal generalized minimal models are compact just like diagonal ordinary minimal models. It was also nice that they are quasi-rational. While we are aware of work you have done in Ref. [35] on generalized non-diagonal minimal models, we are not sure if these allow the same ideas to work and think this is best left for the future.
-
Removed instances of calling $m$ the rank.
-
Clarified statement and added references.
-
It is conceivable that minimal models with finite values of $m$ will also come to play a role in studies such as this (possibly along the lines of section 5.2). So we thought it would be nice to clarify that $m \gg 1$ is not only needed to make fixed point coupling constants small. Our earlier paper could only mention this in a cryptic way due to space constraints.
8, 9, 10, 11. We agree.
-
We have now added definitions for "divergence candidate" and "potential divergence" under Eq. (2.15) and added some commentary under Eq. (2.17) designed to reassure the reader that it involves a very manageable calculation.
-
We have added some discussion under the formula with Hecke operators to explain why it describes a gauging that is anomaly-free.
-
Fixed.
-
Not necessarily but we wanted to err on the side of too much review instead of too little.
-
Indeed, but we were bothered by empty parentheses for the singlet.
-
There are not enough divergences built out of $\phi_{(1,2)}$ but there are enough built out of $\phi_{(1,3)}$. For this reason, we are indeed conjecturing that recombination occurs at the fixed point and we have tried to make this clearer. We also developed slightly the statement for the massive pure $\sigma$ deformation.
-
We included Eq. (3.19) to try to give a flavour for how the calculation changes between sections 3.2 and 3.4.
-
We have switched to a notation more expressive than $n$ and reworded the caption of this table to refer back to section 3. We think this is appropriate now that section 3 has a more satisfactory discussion of degeneracies for each irrep.
-
We have added some explicit currents under Eq. (4.13) in order to explain the simplest example of a redundant degree of freedom that needs to be removed.
21, 22. We agree.
Although you have already abstained from making recommendations about whether to publish, we hope you consider these changes to be an improvement.

Author: Antonio Antunes on 2025-04-02 [id 5329]
(in reply to Report 2 by Jiaxin Qiao on 2025-03-27)Dear Jiaxin,
Thank you for reading the paper carefully and pointing out areas where it can be improved. We have uploaded a new version to the arxiv (v4) in an effort to incorporate your suggestions. Specific changes for the major issues are:
Indeed, explicitly adding the relation between the recombination coefficient $b$ and the OPE coefficients $C$ makes section 2.2 easier to follow and we have now done so.
As you point out, the coefficients in the operators change with N but the number of them (beyond a critical N) does not. It is therefore perfectly possible to get results that are uniform in N and our previous paper carried this out for some low spins. We have now added notes at the beginning and end of section 4 which explain this.
We have found the rank exactly using rational numbers which means one does not have to worry about the stability of the algorithm (which was the one in sympy). Now that this is stated at the end of section 4.2, we believe the rest of this section is sufficient to explain how one goes from operators to the rank of a matrix.
For the minor issues:
Fixed.
The distinction is indeed hard to notice so we added a footnote about that.
We tried to only say "spin up to 10" or $J \leq 10$ when referring to the previous paper because only $J \leq 9$ has been checked here. But this is indeed confusing since tables go up to 10, as you say, and there is only one line (on page 25) about $J = 10$ being too punishing for some irreps. We have tried to improve the situation by adding comments to tables 2 and 8 which warn that spin 10 has not been checked.
We also revisited a couple of Sylvain's points and made further changes listed below. 1. We mentioned the use of refined partition functions in the introduction following the changes in the third version of the paper on arxiv. 2. We have uniformized the nomenclature "divergence candidates". 3. We added a few sentences in the outlook about coupling D-series minimal models.
Jiaxin Qiao on 2025-04-02 [id 5330]
(in reply to Antonio Antunes on 2025-04-02 [id 5329])Hi Antonio,
Thanks for the reply. In version 4, I am confused about the footnote 18 added at the beginning of section 4. I thought for spin 10, the critical value of N should be at least greater than 7 according to (4.10). Why is it equal to 5?
Anonymous on 2025-04-02 [id 5332]
(in reply to Jiaxin Qiao on 2025-04-02 [id 5330])Hi Jiaxin,
Thanks. This was a typo: it is indeed 7, we will fix it.

---

## Round 2 · Referee Report · Jiaxin Qiao (Referee 2) · 2025-3-27

Report
Understanding the landscape of 2d CFTs remains an important open problem. In particular, a concrete example of a 2d CFT satisfying the following properties is still unknown: (a) central charge $c > 1$, (b) unitarity, (c) a normalizable vacuum, (d) modular invariance, and (e) absence of extra continuous global symmetries.
In a previous work [arXiv:2211.16503], the authors proposed a construction that could potentially yield such a 2d CFT. The setup begins with a UV CFT given by $N$ copies of a unitary diagonal minimal model: $\left(\mathcal{M}_{m,m+1}\right)^{\otimes N}$, where $N$ is finite and $m$ is taken to be very large. In this regime, the authors identify two weakly relevant operators, denoted $\sigma$ and $\epsilon$, that preserve the $S_N$ global symmetry and do not trigger other relevant operators along the RG flow. By fine tuning their couplings, they discover two new fixed points, ${\rm FP}_{\pm}^*$. These fixed points are expected to satisfy conditions (a)-(d), but whether condition (e) holds remains unclear. The challenge lies in the fact that the UV theory has an extended chiral symmetry given by $Vir^{\otimes N}$, the product of $N$ Virasoro algebras. It is not known whether $Vir^{\otimes N}$ breaks to the diagonal Virasoro algebra $\widehat{Vir}$ in the IR. This is the main question addressed in the present paper.
The main conjecture of this work is as follows. Fix $N > 4$, and let $T$ be any chiral current operator in the UV theory $\left(\mathcal{M}_{m,m+1}\right)^{\otimes N}$ that transforms as a primary under $\widehat{Vir}$. Then, in the IR fixed point ${\rm FP}_{\pm}^*$, $T$ acquires an anomalous dimension of the form
Instead of performing a two-loop computation of $\gamma_T$, the authors invoke the method of weakly broken symmetries: $\bar{\partial}T = \frac{#}{m} V+O(m^{-2})$, which relates the non-zero leading anomalous dimension to the non-vanishing of the three-point function $\langle T V \sigma\rangle$, where $T$ is the UV current, $V$ is the candidate divergence operator, and $\sigma$ is the relevant deformation operator.
For each spin $J$, there are many chiral currents $T^K$ and candidate divergences $V^L$. The statement that all such currents acquire non-zero anomalous dimensions at order $O(m^{-2})$ is equivalent to that the matrix $M^{LK} = \langle T^L V^K \sigma\rangle$ has maximal rank---equal to the number of independent spin-$J$ currents $T^L$. By first studying the decomposition of the characters of $Vir^{\otimes N}$ into characters of $\widehat{Vir}\times S_N$, the authors showed that for each spin (less than certain truncation) and for each irreducible representation of $S_N$, there are more candidate divergences than currents, making maximal rank possible. They then built up an algorithm for choosing bases of currents and divergences according to $\widehat{Vir}\times S_N$, and verify numerically that maximal rank is indeed achieved for $J \leqslant 10$ and $N = 5,6,7$. This provides a positive test of the conjecture.
This work is a valuable and well-written contribution to the study of irrational 2d CFTs. Based on version 3 of the draft, I have the following suggestions to improve clarity and completeness:
Main Issues 1. Although it is explained in words that the integral in eq.(2.17) is trivial, it would be more reader-friendly to explicitly state the final result. For example, one could write something like: \begin{equation} b^K_L = \pi C_{T^K V^L \sigma} + O(1/m), \end{equation} where $C_{T^K V^L \sigma}$ is the structure constant of the three-point function. This would also make it more natural to begin section 4.2 with eq.(4.16).
-
What is the technical obstruction to extending the check to $J\leqslant10$ and general (but large) values of $N$? According to table 8, there are only finitely many independent currents with $J \leqslant10$.
-
Since the main technical analysis in section 4.2 is performed numerically, it would be helpful to include a brief summary of the algorithm used, before the end of that section.
Minor Issues 1. In table 8, the entry for $J = 2$ is missing in the list of $d_{J,0}^{(N-1,1)}$.
-
Around (4.1), it would be helpful to remark that "$k$" is the subscript of $n$ instead of $L$ (since both $n$ and $k$ are small). I was personally confused about the placement of the subscripts: $L_{-n_{j,k}}$ vs $L_{-n_j,k}$.
-
The last paragraph of section 4 states that the numerical check was done for $J < 10$, but all the tables seem to include preparations for $J = 10$ as well (also the introduction says $J\leqslant10$). Has the case $J = 10$ actually been checked?
Recommendation
Publish (surpasses expectations and criteria for this Journal; among top 10%)

---

## Round 4 · Referee Report · Sylvain Ribault (Referee 1) · 2025-4-3

Report

The authors have made significant improvements, both in substance and in clarity. These include the introduction of refined partition functions, which could be of independent interest. I have no further suggestions.

As a matter of principle I do not make recommendations on whether to publish articles or not. Editorial decisions belong to editors. Since the computer system makes it necessary to have a recommendation, I picked the first one in the list.

Recommendation

Publish (surpasses expectations and criteria for this Journal; among top 10%)

---

## Round 4 · Author Response

Please see our responses to the referees.

---

## Editorial Decision

published